# Downsizing the molecular spring of the giant protein titin reveals that skeletal muscle titin determines passive stiffness and drives longitudinal hypertrophy

**Ambjorn Brynnel, Yaeren Hernandez, Balazs Kiss, Johan Lindqvist, Maya Adler, Justin Kolb, Robbert van der Pijl, Jochen Gohlke, Joshua Strom, John Smith, Coen Ottenheijm, Henk L Granzier***

Department of Cellular and Molecular Medicine, University of Arizona, Tucson, United States

**Abstract** Titin, the largest protein known, forms an elastic myofilament in the striated muscle sarcomere. To establish titin's contribution to skeletal muscle passive stiffness, relative to that of the extracellular matrix, a mouse model was created in which titin's molecular spring region was shortened by deleting 47 exons, the $Ttn^{\Delta 112-158}$ model. RNA sequencing and super-resolution microscopy predicts a much stiffer titin molecule. Mechanical studies with this novel mouse model support that titin is the main determinant of skeletal muscle passive stiffness. Unexpectedly, the in vivo sarcomere length working range was shifted to shorter lengths in $Ttn^{\Delta 112-158}$ mice, due to a ~ 30% increase in the number of sarcomeres in series (longitudinal hypertrophy). The expected effect of this shift on active force generation was minimized through a shortening of thin filaments that was discovered in $Ttn^{\Delta 112-158}$ mice. Thus, skeletal muscle titin is the dominant determinant of physiological passive stiffness and drives longitudinal hypertrophy.
**Editorial note:** This article has been through an editorial process in which the authors decide how to respond to the issues raised during peer review. The Reviewing Editor's assessment is that all the issues have been addressed (see decision letter).
DOI: https://doi.org/10.7554/eLife.40532.001

*For correspondence:
granzier@email.arizona.edu

**Competing interests:** The authors declare that no competing interests exist.

## Introduction

Striated muscles generate the forces and motions which are essential for life. The contractile unit of the striated muscle is the sarcomere, consisting of three types of myofilaments (*Squire et al., 2017*). The thick filaments are located in the middle of the sarcomere and contain the molecular motor myosin that underlies active force development. The thick filaments overlap with the actin-based thin filaments that are attached to the ends of the sarcomere (the Z-disks) and that transmit the active forces generated by myosin (*Squire et al., 2017*). The focus of the present study is on titin, the largest known protein (~3800 kDa), that forms the third myofilament type (*Wang et al., 1984*; *Granzier and Labeit, 2005*), spanning the half-sarcomere from Z-disk to M-band (*Fürst et al., 1988*). The I-band region of titin functions as a molecular spring that extends and generates force (passive force) when muscles are stretched (*Trombitás et al., 1995*). Titin's molecular spring consists of two main sub-segments, the tandem Ig segment (serially-linked immunoglobulin (Ig)-like domains) and the PEVK segment (rich in proline (P), glutamate (E), valine (V), and lysine(K)) (*Trombitás et al., 1998a*; *Bang et al., 2001*). These segments have different biophysical properties and therefore their extensions dominate at different but physiological sarcomere length ranges (*Trombitás et al., 1998b*; *Watanabe et al., 2002a*). Titin's force maintains the structural integrity of the

sarcomere (*Horowits and Podolsky, 1987*) and limits sarcomere length variation along myofibrils (serially-linked sarcomeres) (*Granzier and Pollack, 1990*).

While the contribution of titin to sarcomere passive stiffness (slope of the passive force-sarcomere length relation) in single fibers is widely accepted (*Granzier and Wang, 1993*; *Granzier and Labeit, 2006*; *Prado et al., 2005*), there are conflicting views on the relevance of titin to passive stiffness at the skeletal muscle level. It has been proposed that the extracellular matrix (ECM) dominates whole muscle passive stiffness and that titin is less relevant at the whole muscle level (*Prado et al., 2005*; *Gillies et al., 2011*; *Lieber et al., 2017*; *Tirrell et al., 2012*; *Meyer and Lieber, 2011*). This contrasts with studies that concluded that the passive stiffness of whole muscle fully resides in single fibers (titin) (*Magid and Law, 1985*) and also with work on cardiac muscle, where titin's stiffness exceeds ECM-stiffness (*Granzier and Irving, 1995*; *Wu et al., 2000*). An important goal of the present study was to directly address this controversy. Understanding the role of titin in passive muscle has clinical relevance as titin-based passive stiffness is known to be altered in skeletal muscle diseases. In fascio-scapulohumeral muscular dystrophy (FSHD) titin-based stiffness is increased (*Lassche et al., 2013*) and in cerebral palsy patients, an increase in titin-based passive stiffness has been implicated as well (*Ottenheijm and Granzier, 2010*; *Fridén and Lieber, 2003*). In contrast, diaphragm fibers in chronic obstructive pulmonary disease (COPD) have a reduced titin-based stiffness (*Ottenheijm et al., 2006*). It is unknown if changes in titin-based stiffness carry through to the level of the muscle and, importantly, if altered titin stiffness causes disease, or reflects an adaptation to disease. An in-depth understanding of titin's roles in skeletal muscle stiffness is required.

To aid in this work, a mouse model was created in which titin's stiffness was increased by shortening titin's molecular spring region. We targeted for deletion 47 PEVK exons (*Ttn* exon 112–158), referred to as the $Ttn^{\Delta 112-158}$ model. Transcript, protein and functional studies (muscle mechanics, exercise testing) were performed, to determine the effects of the deletion on the single fiber, whole muscle, and organismal levels. The $Ttn^{\Delta 112-158}$ model also provides a unique opportunity to test the effects of altered titin-based tension on active muscle function. Recently, several novel mechanisms have been proposed that link titin-based tension to active tension. For example, titin might store elastic energy by unfolding Ig domains in passive muscle. Through their refolding during contraction, titin might generate forces that add to the active force (*Eckels et al., 2018*). An alternative mechanism consists of titin-based effects on thick filament structure that contribute to activating the thick filament (*Piazzesi et al., 2018*; *Fusi et al., 2016*).

Studies on the $Ttn^{\Delta 112-158}$ model that was created reveal that in skeletal muscle the size of the PEVK segment length is reduced by ~75%, and that this highly increases the passive stiffness at the sarcomere, single fiber, and whole muscle levels. Furthermore, within the physiological sarcomere length range of skeletal muscle, titin is the dominant determinant of the passive stiffness in both WT and $Ttn^{\Delta 112-158}$ mice. Hypertrophy occurs in $Ttn^{\Delta 112-158}$ mice, due to longitudinal growth that serially adds sarcomeres. This shifts the in vivo working sarcomere length ranges of $Ttn^{\Delta 112-158}$ muscles to shorter lengths, supporting that titin-based stiffness is functionally important and that its level is carefully controlled.

## Results

### Creating the $Ttn^{\Delta 112-158}$ mouse model

The targeting strategy used homologous recombination to replace exons 112–158 (chr2:76,839,202–76,867,333) with the exon numbers from orthologous human titin exons. The 47 targeted exons are all PEVK exons and their location in titin's I-band region is shown in *Figure 1A and B*. These exons are not expressed in the main cardiac titin isoform (*Bang et al., 2001*) and a skeletal-muscle-specific effect is expected. The genotype distribution of mice born to heterozygous parents followed Mendelian genetics (*Figure 1C*) and homozygous $Ttn^{\Delta 112-158}$ mice survived without an outwardly noticeable phenotype (oldest mice ~ 1 year). Homozygous $Ttn^{\Delta 112-158}$ mice had body weights indistinguishable from WT littermates at 60 days of life but as mice grew older a weight reduction appeared with two-way ANOVA measured across all ages revealing a significant effect of genotype on weight (*Figure 1D* top and *Figure 1—figure supplement 1*). The $Ttn^{\Delta 112-158}$ mice had the same skeleton size, as suggested by their tibia lengths that were the same as that of WT mice (*Figure 1D* bottom, *Figure 1—figure supplement 2*). Thus, a viable mouse model was created that had a

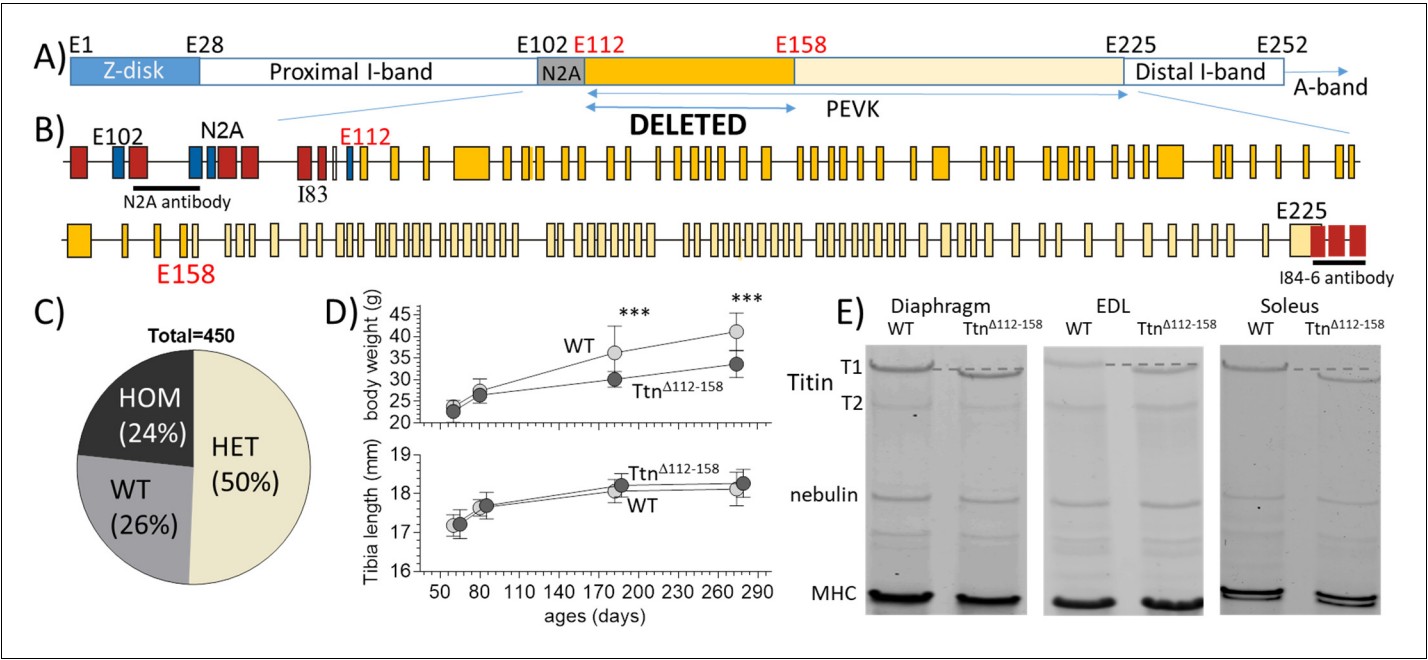

**Figure 1.** The $Ttn^{\Delta112-158}$ mouse model. (**A and B**) Titin's I-band region is shown schematically in (A) and the exon composition of the PEVK and flanking region is indicated in B). The region targeted for deletion is highlighted. (Yellow: PEVK exons; dark yellow: targeted exons; red: Ig domains, blue: unique sequence; the horizontal lines indicate the binding sites of the antibodies used in *Figure 3*). (**C**) Genotype distribution of mice born to heterozygous parents is Mendelian. (**D**) Comparison between WT and homozygous $Ttn^{\Delta112-158}$ mice of body weight (top) and Tibia length (bottom). Two-way ANOVA reveals a significantly reduced BW in $Ttn^{\Delta112-158}$ mice with a multiple comparison analysis revealing a significant reduction at 180 and 275 days. No significant differences in tibia length were found. (All male mice, for female mice, see S1B). (**E**) Protein Gels. Full-length titin (T1) has a higher mobility in $Ttn^{\Delta112-158}$ mice but the mobility of T2 is the same. (T2 is a degradation product of T1 that is present in diaphragm and EDL but absent in soleus). MHC: myosin heavy chain.

DOI: https://doi.org/10.7554/eLife.40532.002

The following figure supplements are available for figure 1:

**Figure supplement 1.** Body weight comparison between WT and homozygous $Ttn^{\Delta112-158}$ female mice.

DOI: https://doi.org/10.7554/eLife.40532.003

**Figure supplement 2.** Tibia length comparison between WT and homozygous $Ttn^{\Delta112-158}$ female mice.

DOI: https://doi.org/10.7554/eLife.40532.004

**Figure supplement 3.** Titin expression analysis.

DOI: https://doi.org/10.7554/eLife.40532.005

normal size but that developed over time a weight deficit. Considering the lack of a weight difference in 60 day old mice, most studies were conducted at this age and with male mice, unless indicated otherwise. Homozygous $Ttn^{\Delta112-158}$ mice were studied and experiments were largely focused on the diaphragm, because of its critical importance for the vital respiratory function, and two contrasting peripheral muscles, the slow-twitch soleus, and fast-twitch EDL muscles.

A protein analysis addressed whether changes in titin expression or titin degradation occurred in $Ttn^{\Delta112-158}$ mice, because if they did, titin-based passive stiffness would be affected. It is well known that skeletal muscle titin appears on gels as two bands, a major upper band which is full-length titin (or T1) and a minor lower band (T2) that is a large degradation product that consists mainly of the A-band segment of titin (*Helmes et al., 1996*). $Ttn^{\Delta112-158}$ muscle lysates also revealed T1 and T2 bands but, compared to WT lysates, T1 had a higher mobility in the $Ttn^{\Delta112-158}$ mouse (*Figure 1E*), as expected from the exon deletion (representing a maximum of 181 kDa protein). The mobility of T2 was unaffected (*Figure 1E*), also as expected because T2 of WT mice does not contain the targeted PEVK segment (*Helmes et al., 1996*). In each muscle type, the total titin (T1 +T2) expression level was determined, relative to myosin heavy chain, and the amount of T1 relative to total titin. No differences were found in titin expression in any of the studied muscle types (*Figure 1—figure supplement 3*).

As titin is known to undergo extensive alternative splicing (*Bang et al., 2001*) it was tested whether in response to deleting exons 112–158, splicing had adapted in the $Ttn^{\Delta 112-158}$ mouse, by up-or-down-regulating exons outside the targeted area. This could occur but not noticeably alter protein mobility on gels because of their limited resolution in the mega-Dalton protein size range. A RNA sequencing study was performed on diaphragm, EDL and soleus muscles and the percent-spliced-in-index (PSI) was determined for each *Ttn* exon. *Figure 2A* shows the diaphragm data and *Figure 2—figure supplements 1–3*, EDL and soleus data. RNA sequencing confirmed that the PEVK exons 112–158 were absent in the $Ttn^{\Delta 112-158}$ transcript, in all three muscle types (*Figure 2B* and *Figure 2—source data 1*). Only minimal differences outside the deleted region were found. In all muscle types, there was only consistent upregulation of Z-disk exons 12 and 13 and PEVK exon 159. These adaptations are not expected to significantly affect titin's passive stiffness (see Discussion).

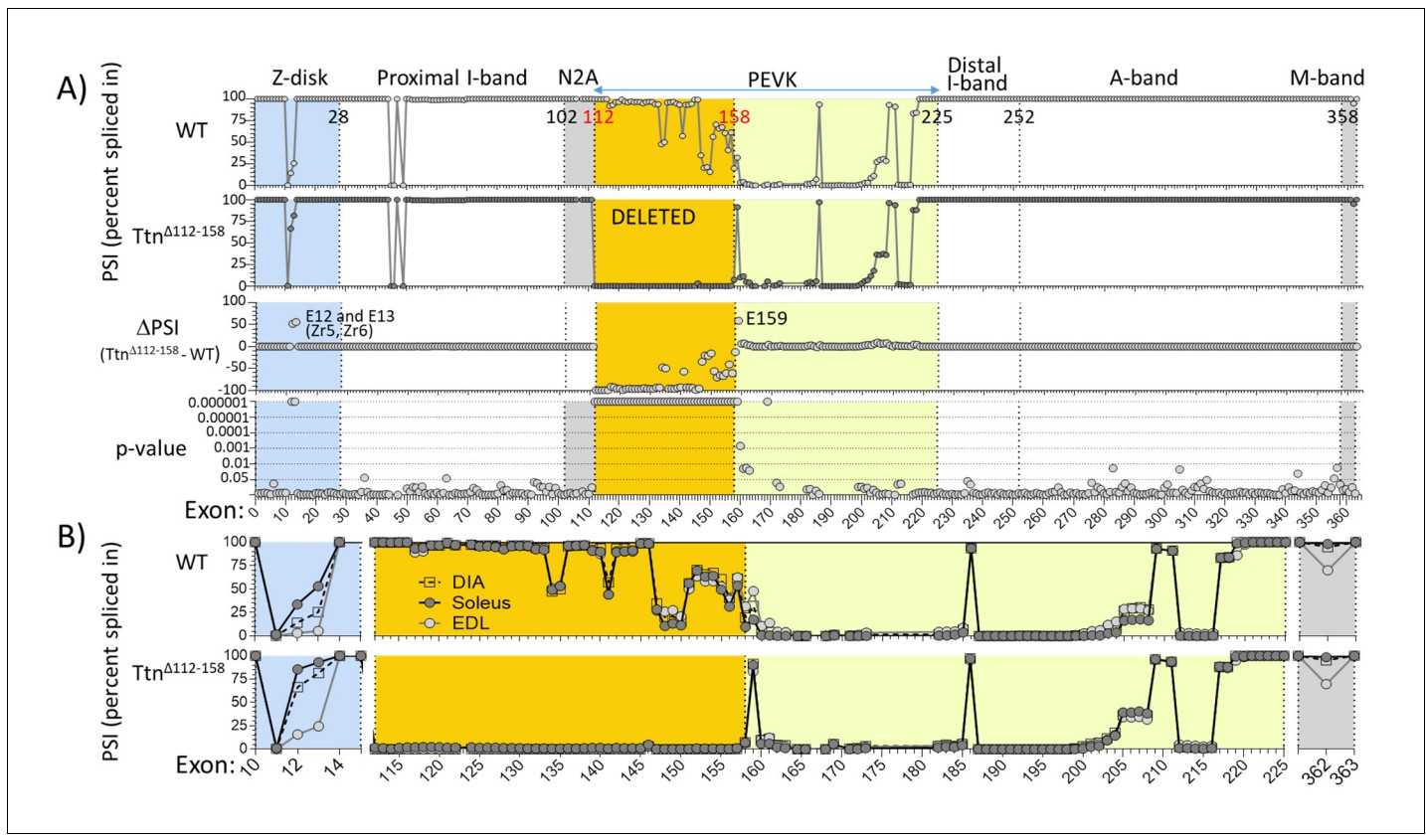

**Figure 2.** Titin exon expression analysis. (A) RNAseq-based titin exon expression of diaphragm muscle, expressed as percent spliced-in (PSI) in WT and $Ttn^{\Delta 112-158}$ mice (top and below), and the PSI difference between the two genotypes and the p-value of PSI difference (bottom and above). The targeted exons are absent in the $Ttn^{\Delta 112-158}$ mouse with minimal changes in the Z-disk and M-band. (p-value scale is –log(p-value) with values > 6 shown as 6.) (B) PSI of Diaphragm, EDL and soleus muscle of WT (top) and $Ttn^{\Delta 112-158}$ (bottom) mice. Only shown are areas within the gene where PSI differences exist between the genotypes. Z-disk (blue), PEVK (yellow) and M-band (gray). Zr: Z repeat.

DOI: https://doi.org/10.7554/eLife.40532.006

The following source data and figure supplements are available for figure 2:

**Source data 1.** Percent Spliced In source data.
DOI: https://doi.org/10.7554/eLife.40532.010

**Figure supplement 1.** Titin expression analysis in EDL muscle.
DOI: https://doi.org/10.7554/eLife.40532.007

**Figure supplement 2.** Titin expression analysis in soleus muscle.
DOI: https://doi.org/10.7554/eLife.40532.008

**Figure supplement 3.** Titin expression analysis in diaphragm, EDL and soleus muscle, in part of Z-disk, complete PEVK and part of M-band regions.
DOI: https://doi.org/10.7554/eLife.40532.009

PSI values were used to determine the size of the PEVK segment. Many PEVK exons have a PSI value that is intermediate between 100% (the exon is spliced in) or 0% (the exon is skipped), suggesting that different isoforms exist in the muscle sample. The average transcript size was determined by multiplying the measured PSI for each exon by the number of residues encoded by the exon, which resulted in 1611, 1538 and 1613 PEVK residues for WT and 421, 429 and 412 for $Ttn^{\Delta112-158}$ transcript, in diaphragm, EDL and soleus muscle, respectively. It is worthwhile noting how similar the three muscle types are, both in WT (*Figure 2B* top) and $Ttn^{\Delta112-158}$ mice (*Figure 2B* bottom).

In summary, a novel mouse model was created in which 47 exons have been deleted from the PEVK region of titin. Homozygous mice were viable, had minimal adaptations in titin splicing outside the targeted region, and titin protein expression levels were normal. Importantly, the PEVK segment was reduced from ~1600 to~420 residues. The $Ttn^{\Delta112-158}$ model is well-suited to study the effects of shortening titin's spring region on the stiffness of titin and the downstream effects of increased titin-based stiffness on passive and active muscle function.

## PEVK segment extension and titin stiffness

To evaluate the effects of the shortened PEVK segment on titin-based stiffness the extension of the PEVK segment was measured in sarcomeres stretched by varying degrees. For this purpose, super-resolution optical microscopy was used with antibodies that flank the PEVK segment (the N2A antibody and I84-86 antibody, shown in *Figure 1B*). *Figure 3A* shows sample images. In both Diaphragm and EDL muscles of $Ttn^{\Delta112-158}$ mice the extension (z) of the PEVK segment was reduced (*Figure 3B*). The contour length (CL) of the PEVK segment was estimated from the number of amino acids contained in the PEVK segment (see above) and assuming a random coil structure with a maximal residue spacing of 0.38 nm (*Watanabe et al., 2002b*); the relative extension of the PEVK was calculated as z/CL. The obtained values were used in the wormlike chain equation and the force per

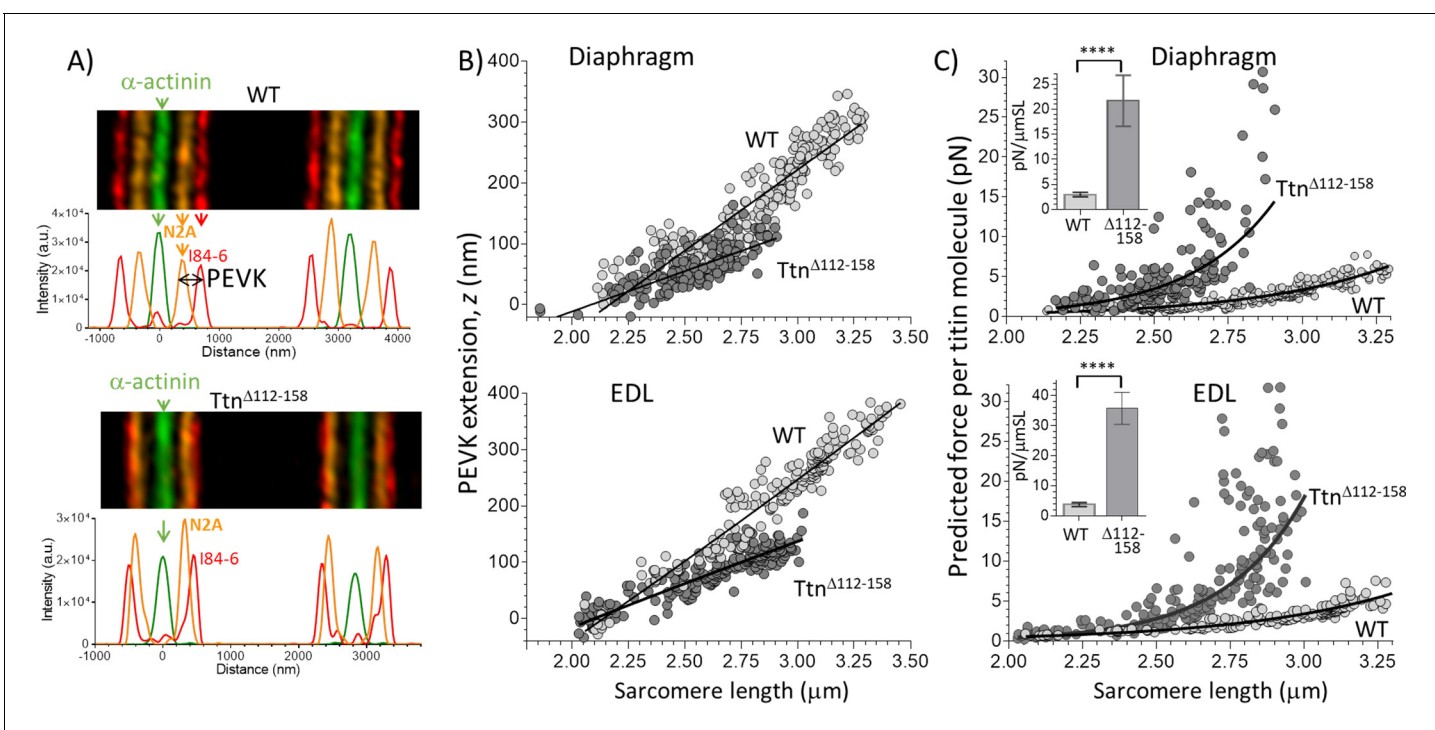

**Figure 3.** Extension of the PEVK segment and predicted titin-based force. (A) Example immuno-fluorescence images and densitometry of sarcomeres labeled with N2A and I84-86 antibodies that flank the PEVK (*Figure 1B* shows binding sites). (B) PEVK extension (z) as function of sarcomere length in muscles stretched to different lengths. (C) Force per titin molecule as a function of sarcomere length. Inset shows the average stiffness of titin in the 2.45–2.75 µm SL range. The measured PEVK extension predicts a large increase in titin-based stiffness in $Ttn^{\Delta112-158}$.
DOI: https://doi.org/10.7554/eLife.40532.011

titin molecule was determined (details in the Method section). The results in *Figure 3C* reveal in diaphragm (top) and EDL (bottom), a much steeper force increase with sarcomere extension in $Ttn^{\Delta 112\text{-}158}$ mice. From these data, the average titin stiffness was determined in the sarcomere length range of 2.45–2.75 μm (this captures most of their physiological SL range, see also Discussion). The insets of *Figure 3C* reveal that, compared to WT, the titin-based stiffness of $Ttn^{\Delta 112\text{-}158}$ mice is predicted to be greatly increased.

## Passive stiffness of muscle

To investigate the effect of shortening titin's PEVK segment on passive muscle stiffness, the passive tension - sarcomere length relation was studied at the level of the whole muscle (details in Materials and methods). When the passive tension measurements were completed, the muscle was chemically fixed at its slack length, fiber bundles were dissected and sarcomere length (SL) was measured. From this SL value, the SL at each muscle length during the mechanics experiment was calculated (see Materials and methods). The obtained passive tension- SL relations are given in *Figure 4*. In all three muscle types, passive tension increased much more steeply with SL in the $Ttn^{\Delta 112\text{-}158}$ muscles. From these data, the average passive stiffness (slope of tension – sarcomere length relation) in the sarcomere length range 2.45–2.75 μm was calculated. The results showed a several-fold increased passive stiffness in $Ttn^{\Delta 112\text{-}158}$ muscles (insets of *Figure 4*).

Titin is not the only source of muscle passive tension, the extracellular matrix (ECM) also contributes (*Lieber et al., 2017*; *Huijing, 1999*). To determine how much each contributes to total passive muscle tension, the EDL fifth toe muscle (one of the heads of the EDL) was studied because it is thin enough to: (1) measure sarcomere length with laser diffraction in intact muscle, (2) fully demembranate the muscle ('skinning') with detergent, and (3) extract the skinned muscle with KCl/KI solutions that remove titin's anchors in the sarcomere, leaving the ECM behind but abolishing titin-based tension (see Materials and methods for technical details and Discussion for scientific background).

Passive tensions measured before and after skinning reveal no skinning-induced difference in passive tension, a result found in both genotypes (*Figure 5A and B*). There is a significant though minor stiffness *increase* in the WT muscles in the 2.45–2.75 μm sarcomere length range (*Figure 5A*, inset). The increase is small and, furthermore, no difference occurs in $Ttn^{\Delta 112\text{-}158}$ muscle (*Figure 5B*, inset). Thus, skinning had little effect on the passive tension of muscle.

To validate the KCl/KI extraction method and test whether it only abolishes titin-based tension, a study on mechanically skinned single <u>fibers</u> (from the EDL 5th toe muscle) was also performed. In these fibers the ECM is absent, and the passive tensions are only titin-based. KCl/KI extraction greatly lowered passive tension of both WT and $Ttn^{\Delta 112\text{-}158}$ skinned whole EDL 5th toe muscle (*Figure 5A and B*) and the KCl/KI sensitive tension was similar in magnitude to that of single fibers, as was the stiffness in the 2.45–2.75 μm sarcomere length range (*Figure 5C*). Thus, it is likely that the extraction sensitive tension of skinned whole muscle is titin-based and that the extraction-insensitive tension is ECM-based.

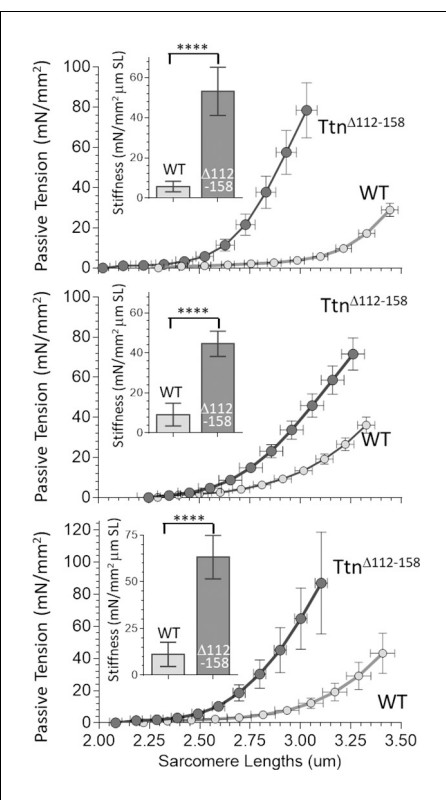

**Figure 4.** Passive Muscle stiffness. The passive tension – sarcomere length relation was determined in Diaphragm (top), EDL (middle) and soleus (bottom) intact muscle from WT and $Ttn^{\Delta 112\text{-}158}$ mice. Passive tension is highly increased in $Ttn^{\Delta 112\text{-}158}$. The insets show average passive stiffness in the 2.45–2.75 μm SL range. Passive stiffness is increased ~5–10 fold in $Ttn^{\Delta 112\text{-}158}$ mice.

DOI: https://doi.org/10.7554/eLife.40532.012

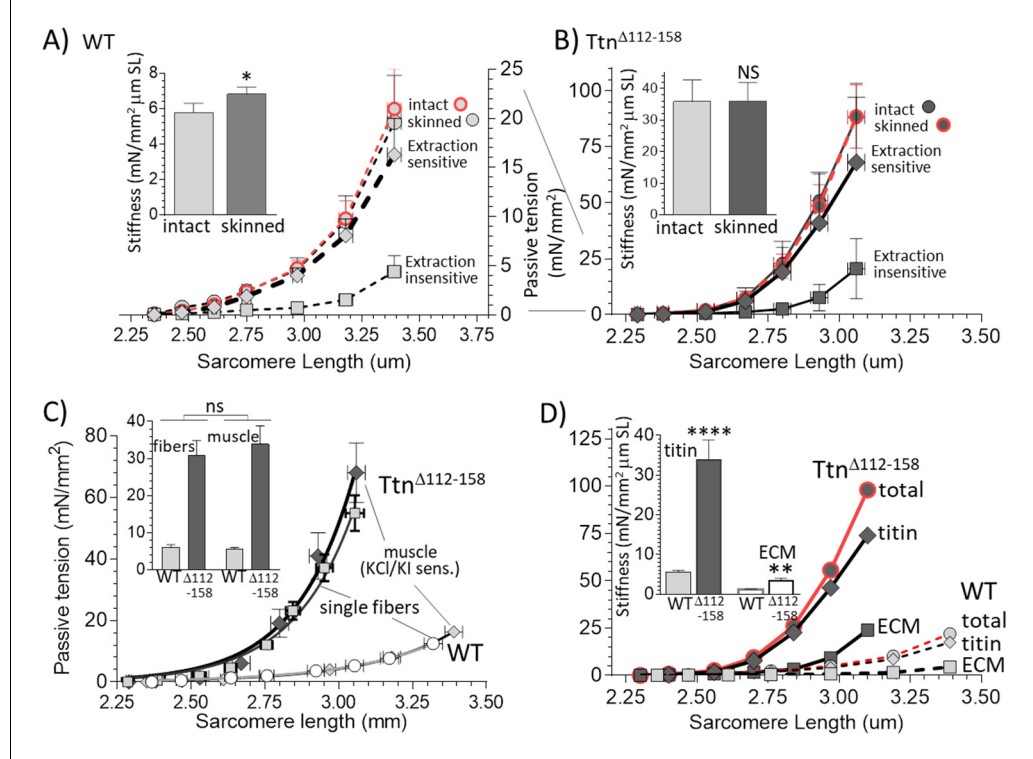

**Figure 5.** Functionally dissecting passive muscle stiffness. (**A and B**) Passive tension in EDL muscle (5th toe muscle) was measured in intact muscle, then again after skinning, and a final time after KCl/KI extraction (extraction insensitive tension). The tension is minimally affected by skinning and tensions before and after skinning largely overlap. The difference between before and after KCl/KI extraction is the extraction-sensitive tension. (**A**) WT, (**B**) $Ttn^{\Delta112-158}$ muscles (note ~4 fold different vertical scales). Insets of A and B show passive stiffness of intact and skinned muscle (SL range 2.45–2.75 μm). Skinning minimally impacts passive stiffness. (**C**) Comparison of extraction sensitive muscle tension with the passive tension of mechanically skinned single EDL fibers. The inset shows the average stiffness (SL range 2.45–2.75 μm) for single fibers (left two bars) and whole muscle (right two bars). Results are not significantly different, supporting that the extraction-sensitive tension is titin-based (see text for details). (**D**) Overlay of WT and $Ttn^{\Delta112-158}$ skinned muscle total tension, titin-based tension and ECM-based tension. The inset shows the stiffness (SL 2.45–2.75 μm) for titin and ECM stiffness. Both titin and ECM have increased stiffness in $Ttn^{\Delta112-158}$ muscles, but the effect is much larger for titin.

DOI: https://doi.org/10.7554/eLife.40532.013

The titin-based and ECM-based tension of EDL 5th toe muscle were both significantly higher in $Ttn^{\Delta112-158}$ muscle (**Figure 5D**). The titin-based passive stiffness in the 2.45–2.75 μm sarcomere length range was increased ~5 fold (**Figure 5D**, inset). The ECM stiffness was increased as well but its magnitude remained much below the stiffness of titin. Thus, passive $Ttn^{\Delta112-158}$ skeletal muscles are much stiffer, which largely can be accounted for by increased titin stiffness.

## Active tension

To determine whether the increased titin-based tension of $Ttn^{\Delta112-158}$ muscles affected *active* tension, intact diaphragm, EDL (whole muscle and 5th toe muscle) and soleus muscles were electrically stimulated at $L_0$ (the optimal length for force generation), using a range of frequencies until a maximal tetanic tension was generated (see Materials and methods for details). Passive tension in $Ttn^{\Delta112-158}$ muscle was much higher at $L_0$ (for example, in EDL 5th toe muscle, 11.8 ± 1.1 compared to 2.5 ± 0.4 mN/mm2 in WT muscle). An example of an active-tension data set is in **Figure 6A**. The WT and $Ttn^{\Delta112-158}$ curves largely overlapped. The maximal active tensions for all studied muscle types are in **Figure 6B**. The different muscle types generated different active tension levels in WT animals, as expected because these muscles express different myosin isoforms that impact force

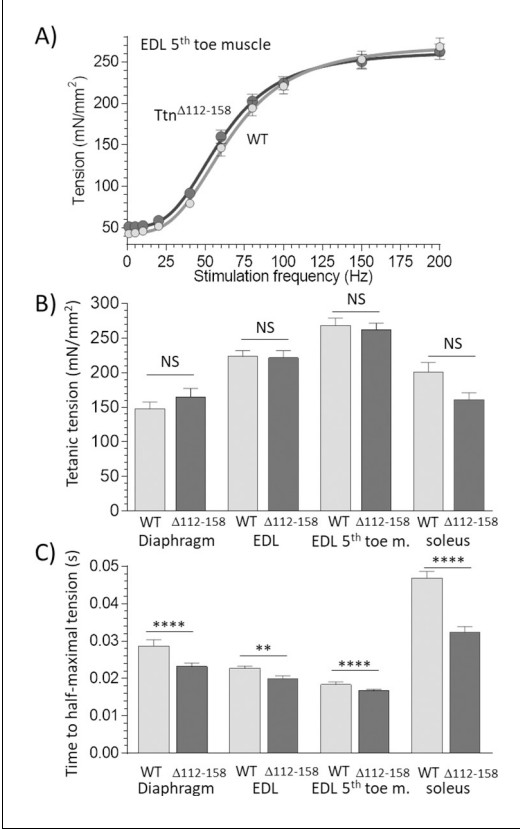

**Figure 6.** Active tension. (**A**) Example results of tetanic tension vs. stimulation frequency. Results of EDL 5th toe muscle are shown. Results of WT and $Ttn^{\Delta112-158}$ muscles closely overlap. (**B**) Maximal tetanic tension in Diaphragm (150 Hz), whole EDL muscle (200 Hz), 5th toe muscle of the EDL (200 Hz) and soleus muscle (150 Hz). Two-way ANOVA reveals that there is no significant difference in maximal tetanic tension in $Ttn^{\Delta112-158}$ muscles with a multiple comparison analysis revealing also no effects in individual muscle types. (**C**) Time to reach half-maximal active tetanic tension is significantly altered in $Ttn^{\Delta112-158}$ muscles (two-way ANOVA) with a multiple comparison analysis revealing a significant reduction in all $Ttn^{\Delta112-158}$ muscle types.
DOI: https://doi.org/10.7554/eLife.40532.014

The following figure supplement is available for figure 6:

**Figure supplement 1.** Myosin Heavy Chain (MHC) isoform analysis.
DOI: https://doi.org/10.7554/eLife.40532.015

generation (*Schiaffino and Reggiani, 2011*). However, no genotype effect on maximal active tension was found (*Figure 6B*). The speed of force development was also studied, by measuring the time required to generate half-maximal tetanic tension. A significant genotype effect was found, and a multiple comparison analysis revealed a significantly reduced time to half-maximal tension in all muscle types of $Ttn^{\Delta112-158}$ mice (*Figure 6C*). To evaluate whether this can be explained by a fiber type switch in $Ttn^{\Delta112-158}$ mice, a myosin-isoform analysis was performed in the 3 muscle types but no genotype effect was found (*Figure 6—figure supplement 1*). Overall, the studies on activated muscles support that force development is faster in $Ttn^{\Delta112-158}$ mice, but that maximal active tension levels are unaffected.

## Exercise performance

We anticipated that the movement range would be negatively impacted by the highly increased passive muscle stiffness in $Ttn^{\Delta112-158}$ mice and that this would reduce exercise performance. This was tested in a free-wheel running study in which mice ran voluntarily and the running distance and running speed were continuously monitored for 40 days. However, the average running speed and the average running distance (per 24 hr) of $Ttn^{\Delta112-158}$ mice were not different from WT mice (*Figure 7A*). Thus, in a voluntary running test, the $Ttn^{\Delta112-158}$ mice are not limited by their stiffer muscles. A metabolic treadmill study was also performed in which both the incline and speed of the treadmill were progressively increased until the mouse reached its physical limit. Even when mice were pushed to their extreme, no genotype effect was found in the maximal running speed and the total distance covered (*Figure 7B*, left and middle). The oxygen consumption (VO2) at the maximal running speed was also not different (*Figure 7B*, right). It is generally assumed that increased passive stiffness (as occurs in the elderly) negatively affects muscle function and the obtained results suggest that either passive muscle stiffness does not affect exercise performance or, alternatively, that the mice have successfully adapted to the stiffer titin.

## Physiological sarcomere length working range

One possibility for adapting to stiffer passive muscles is to operate at shorter sarcomere lengths, where operating passive muscle stiffness is less. To test this, a whole-body perfusion-fixation study was performed, in which muscles were chemically fixed in a known physiological state and their sarcomere lengths measured ex vivo in dissected muscle strips. Before perfusion, ultrasound imaging

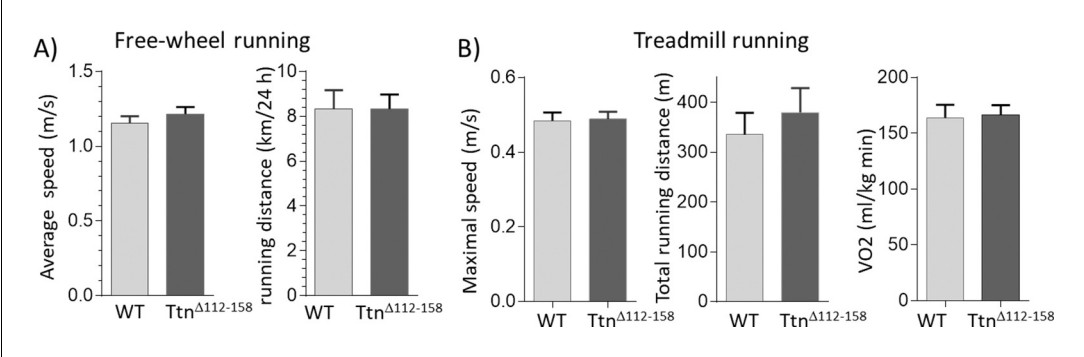

**Figure 7.** Exercise capacity. (**A**) Free-wheel running exercise. Left, average running speed; Right, average distance ran in a 24 hr period. There is no significant difference in running speed and running between $Ttn^{\Delta112-158}$ and WT mice. (**B**) Treadmill running exercise in which the mice had to run uphill and at a progressively increasing speed until they failed to keep up (see Materials and methods). Maximal running speed total running distance and VO2 consumption at maximal running speed (normalized to body weight) revealed no genotype effects. (Female mice were used.).
DOI: https://doi.org/10.7554/eLife.40532.016

was used to measure during the respiration cycle, the diaphragm strain amplitude (fractional shortening of the muscle fibers during inspiration). Diaphragm strain amplitude did not vary between $Ttn^{\Delta112-158}$ and WT mice (*Figure 8A*). The ultrasound study also revealed that during perfusion fixation the diaphragm arrested at its expiration state (the passive state of the diaphragm). The measured sarcomere length of chemically fixed muscle, therefore, reflects the physiological sarcomere length limit and the measured strain reduction during inspiration allows the minimal sarcomere length during contraction to be calculated. Results reveal that the $Ttn^{\Delta112-158}$ mice had a significantly reduced sarcomere length both during expiration and inspiration (*Figure 8B*); the physiological sarcomere length range of the diaphragm was 2.85 µm to 2.22 µm in WT mice but only 2.37 µm to 1.88 µm in $Ttn^{\Delta112-158}$ mice. To determine whether this adaptation is unique to the diaphragm or also occurred in peripheral muscles, the following experiments were performed. Prior to perfusion-fixation one of the hind legs was gently bent in a fully plantarflexed position and the other hind leg in a fully dorsiflexed position (see Materials and methods). Following perfusion fixation, the EDL and soleus muscles were dissected and their sarcomere lengths determined. *Figure 8C* reveals in $Ttn^{\Delta112-158}$ mice shorter sarcomere lengths in stretched as well as shortened EDL and soleus muscles. Overall, the results (schematically shown in *Figure 8D*) support that muscles of the $Ttn^{\Delta112-158}$ mice have shifted their physiological sarcomere lengths to shorter lengths, in the diaphragm as well as peripheral muscles. Hence, within their in vivo working range, operating passive stiffness of muscles in $Ttn^{\Delta112-158}$ mice will be much less than it would have been if they had the same working range as WT mice (see also Discussion).

## The number of sarcomeres in series and longitudinal muscle growth

To explore the mechanisms underlying the shorter sarcomere length working range, single fibers were dissected from end-to-end in chemically fixed muscles (Materials and methods). Fiber lengths in the $Ttn^{\Delta112-158}$ mice were significantly increased by 13.5% (*Figure 8—figure supplement 1*). The sarcomere length of the dissected fibers was also determined (laser diffraction) and by dividing the measured fiber length by the obtained sarcomere length, the total number of sarcomeres along the fibers was calculated. This revealed a significant and large increase (~30%) in the number of sarcomeres in all examined $Ttn^{\Delta112-158}$ muscles (*Figure 9*).

To determine whether the increased number of sarcomeres resulted in increased muscle weights, a range of muscle types were studied, including the diaphragm, EDL and soleus muscles. Muscle weights were determined at 60, 80, 183 (6 mo) and 274 (9 mo) days of life. Most $Ttn^{\Delta112-158}$ muscle types had a significantly increased muscle weight, relative to WT controls (*Figure 10A–C* and *Figure 10—figure supplement 1*). The increased muscles weights, combined with the reduced body weight and normal body size (revealed by the normal tibia lengths, discussed above), suggests that the $Ttn^{\Delta112-158}$ mice are likely to have a lower body fat content. A two-way ANOVA analysis with age and genotype as factors revealed that in most muscle types, both genotype and age had a

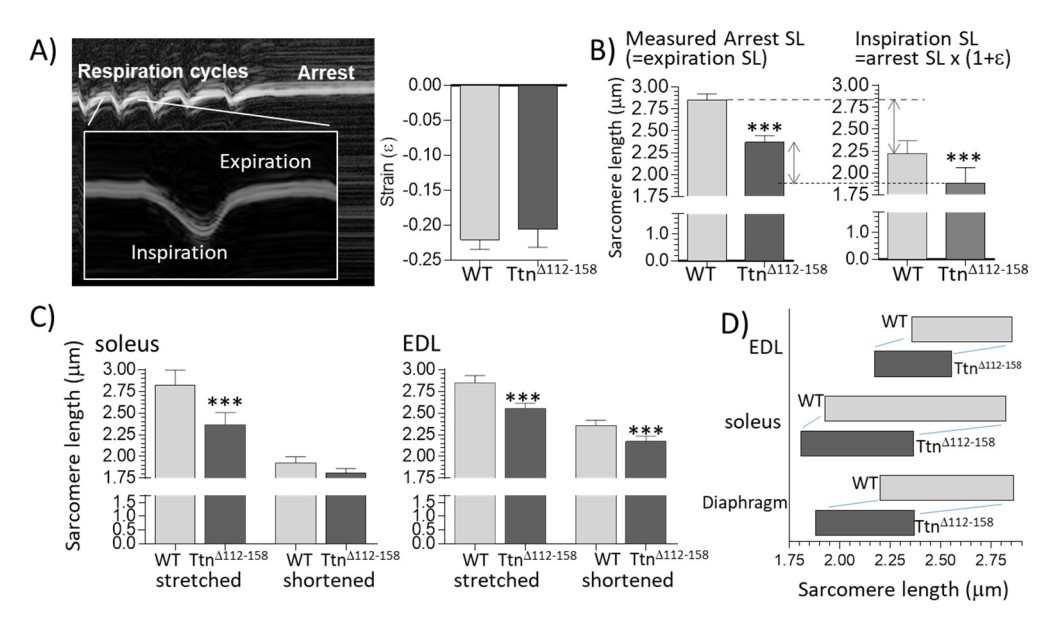

**Figure 8.** Sarcomere length working range. (**A**) Motion-mode Echo image of the diaphragm in a WT mouse. The change in strain was measuring during inspiration (shortening) and then the mouse was perfusion-fixed which arrested the diaphragm in the expiration state. (**A**) Right: Strain amplitude (ε) measured on echo images during inspiration is the same in WT and $Ttn^{\Delta112-158}$ mice. (Strain amplitude is defined as the fractional muscle shortening during inspiration.) (**B**) Left: measured sarcomere length (SL) in muscle strips dissected from the perfusion-fixed diaphragm is much less in $Ttn^{\Delta112-158}$ than in WT mice. Right: calculated SL at end-inspiration, determined from the measured arrest SL and strain amplitude. (**C**) Measured sarcomere length in perfusion-fixed soleus muscle (left) and EDL muscle (right) in mice in which one hind leg was in a fully plantarflexed position while the other hind leg was held in a fully dorsiflexed position. (**D**) Sarcomere working range. The working range of all examined muscle types is reduced in $Ttn^{\Delta112-158}$ mice. (Right side of the box is the maximal length and the left side the minimal length).

DOI: https://doi.org/10.7554/eLife.40532.017

The following figure supplement is available for figure 8:

**Figure supplement 1.** Fiber lengths and muscle lengths are increased in $Ttn^{\Delta112-158}$ mice.

DOI: https://doi.org/10.7554/eLife.40532.018

---

significant effect on muscle weight but with no significant interaction between genotype and age. This indicates that age similarly affected muscle weights of WT and $Ttn^{\Delta112-158}$ mice. The muscle weight increase of $Ttn^{\Delta112-158}$ muscle averaged across all ages varied from 5% for the diaphragm to 13% in the quadriceps and the average increase of all muscles was $9.0 \pm 1.0\%$. Since the cross-sectional area of muscles was not increased (**Figure 10D**) it is likely that the increased weight in $Ttn^{\Delta112-158}$ muscles is largely accounted for by longitudinal muscle growth.

## Force-sarcomere length relation

Although decreasing the physiological sarcomere lengths of the $Ttn^{\Delta112-158}$ mice lowers the operating passive muscle stiffness, this adaptation is expected to impact active tension as well. Shorter sarcomeres will have altered overlap between thin and thick filaments and this will affect force via the force-sarcomere length relation. This relation has a plateau at sarcomere lengths where thin-thick filament overlap is maximal (thin filament tips are located in the bare-zone of the thick filaments) and at longer sarcomere lengths force is reduced due to partial overlap between thin and thick filaments (descending limb) and at shorter sarcomere lengths force is reduced due to double overlap of titin filaments (ascending limb), see **Figure 10B**, bottom. To gain insights into how a reduction in operating physiological sarcomere length range affects active force generation, the theoretical force-sarcomere length relation of the diaphragm was determined. The A-band length is known to be a constant in different skeletal muscle types (1.6 μm) (**Sosa et al., 1994**), but the thin filament length

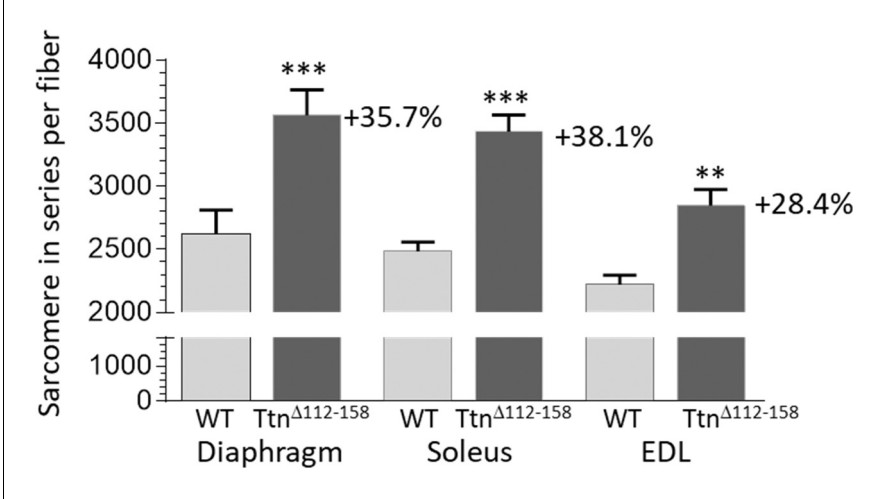

**Figure 9.** Sarcomere in series per muscle fiber. Two-way ANOVA reveals a significantly increased number of sarcomeres in single fibers in all studied muscle types in $Ttn^{\Delta112-158}$ mice. The average increase in the number of sarcomeres in series in all muscle types is 34%.

DOI: https://doi.org/10.7554/eLife.40532.019

varies and has been speculated to respond to the working sarcomere length range (*Littlefield and Fowler, 2008*). The thin filament length was therefore measured, by localizing the thin filament capping protein Tmod4 in diaphragm muscle (Materials and methods). The thin filament length was significantly reduced (*Figure 11A*), from 1165 nm in WT to 1102 nm in $Ttn^{\Delta112-158}$ mice.

The experimentally determined thin filament length and the 1.6 µm thick filament length and 0.15 µm bare-zone width value from the literature (*Sosa et al., 1994*) allows the construction of the force-SL relation in WT and $Ttn^{\Delta112-158}$ diaphragm (for details, see Materials and methods). Results are shown in *Figure 11B*. Due to their shorter thin filaments, the $Ttn^{\Delta112-158}$ F-SL curve (red) is left-shifted relative to the WT curve (dark grey). The experimentally determined sarcomere length working ranges are shown by broken lines. The WT mice operate on the descending limb and part of the plateau, but in contrast, the $Ttn^{\Delta112-158}$ muscle starts out on the plateau and shortens on the ascending limb. The average force within the working range was calculated (using the predicted forces at 0.05 µm SL increments). Within the working range, WT sarcomeres developed on average 91% of the maximal force and $Ttn^{\Delta112-158}$ sarcomeres 96% of the maximal force. Thus, although the reduction in the physiological sarcomere length working range is quite large in $Ttn^{\Delta112-158}$ mice, due to the reduced thin filament length there appears to be little downside for active force development.

## Discussion

### The $Ttn^{\Delta112-158}$ model

The PEVK segment is one of the two main extensible segments in the molecular spring region of skeletal muscle titin (*Trombitás et al., 1998a*). By deleting exons 112–158 from the *Ttn* gene, a mouse model was created in which skeletal muscles express a ~75% shortened PEVK segment (~420 vs~1600 residues). As expected from the much shortened PEVK segment, the passive stiffness of single muscle fibers is greatly increased in $Ttn^{\Delta112-158}$ mice (*Figure 5C*), in line with the prediction based on the reduced PEVK segment extension in stretched sarcomeres (*Figure 3*). These results support the importance of the PEVK segment in determining the passive stiffness of skeletal muscle fibers.

Most of the PEVK exons that were deleted are fully expressed in WT skeletal muscle (PSI ~ 100%) but 15 exons are expressed at a reduced level (*Figure 2B*), 10 have a PSI of ~50% and five exons have a PSI of ~25%. This indicates the presence of multiple splice variants in the samples. In addition to the expected absence of the 47 targeted PEVK exons in $Ttn^{\Delta112-158}$ mice, only minimal adaptions elsewhere in the titin transcript were found. One adaptation that was consistently detected was in

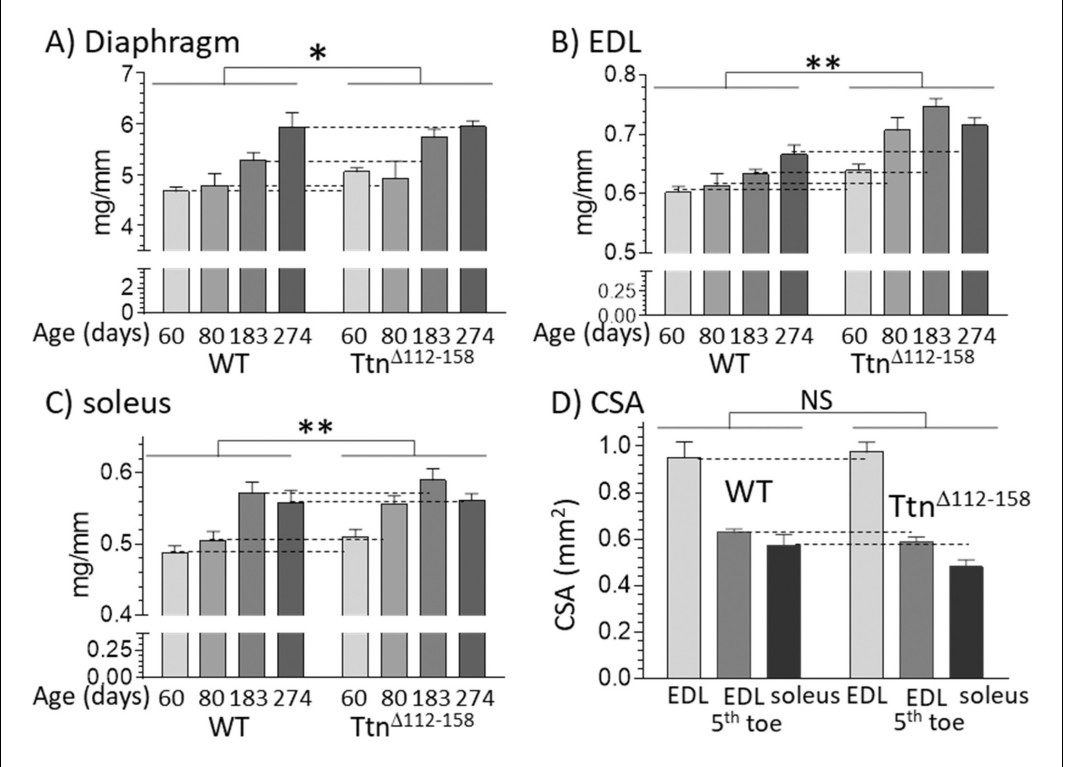

**Figure 10.** Muscles of $Ttn^{\Delta112\text{-}158}$ mice are hypertrophied without an increase in cross-sectional area. (A–C) Hypertrophy is expressed as muscle weight (mg) normalized to tibia length (mm). Two-way ANOVA reveals a significantly increased hypertrophy in all muscle types of the $Ttn^{\Delta112\text{-}158}$ mice (male mice, group sizes, 5–13). (D) Cross-sectional area (CSA) of muscles. CSA was measured in passive muscles held at their slack length in 2 month-old mice (group size 5–9). Two-way ANOVA reveals no significant genotype effect on CSA with a multiple comparison analysis revealing no genotype differences for individual muscle types.
DOI: https://doi.org/10.7554/eLife.40532.020

The following figure supplement is available for figure 10:

**Figure supplement 1.** Muscles of $Ttn^{\Delta112\text{-}158}$ mice are hypertrophied.
DOI: https://doi.org/10.7554/eLife.40532.021

exon 159, immediately downstream from the deleted PEVK region. This exon makes in $Ttn^{\Delta112\text{-}158}$ mice a novel splice junction with exon 112 and as a result exon 159 is upregulated (*Figure 2B*). Exon 159 encodes only 28 PEVK residues and its upregulation will only slightly increase the length of the PEVK segment and minimally lower passive stiffness (this has been accounted for when calculating fractional extensions and sarcomere stiffness). A second consistent adaptation outside the targeted region is the upregulation of the Z-disk exons 12 and 13 (*Figure 2B*). These exons encode ~45 residue repeats that interact with α-actinin and that are known as Z-repeats (*Gautel et al., 1996*). The *Ttn* gene contains seven Z-repeats that have variable expression levels in different muscle types (*Granzier et al., 2007*). It has been proposed that this variable expression of Z-repeats is a means of assembling Z-disks of variable thickness and mechanical strength in different fiber types (*Gautel et al., 1996*; *Sorimachi et al., 1997*). However, considering that there is no fiber type switch in the muscles that were studied in $Ttn^{\Delta112\text{-}158}$ mice (*Figure 6—figure supplement 1*), it is unlikely that the upregulation of exons 12 and 13 reflects a change towards fiber types with wider Z-disks. Instead, we propose that the upregulation of exons 12 and 13 is triggered by the increased titin-based tension in $Ttn^{\Delta112\text{-}158}$ mice, in order to mechanically strengthen this critical region of the sarcomere when titin's stiffness is high. The similar expression patterns of the PEVK exons in the three studied muscle types (*Figure 2B*) was unexpected, considering the different fiber type composition and functions of the muscle types that were studied (diaphragm, soleus, and EDL). Thus, highly similar splice patterns can exist in distinct muscle types.

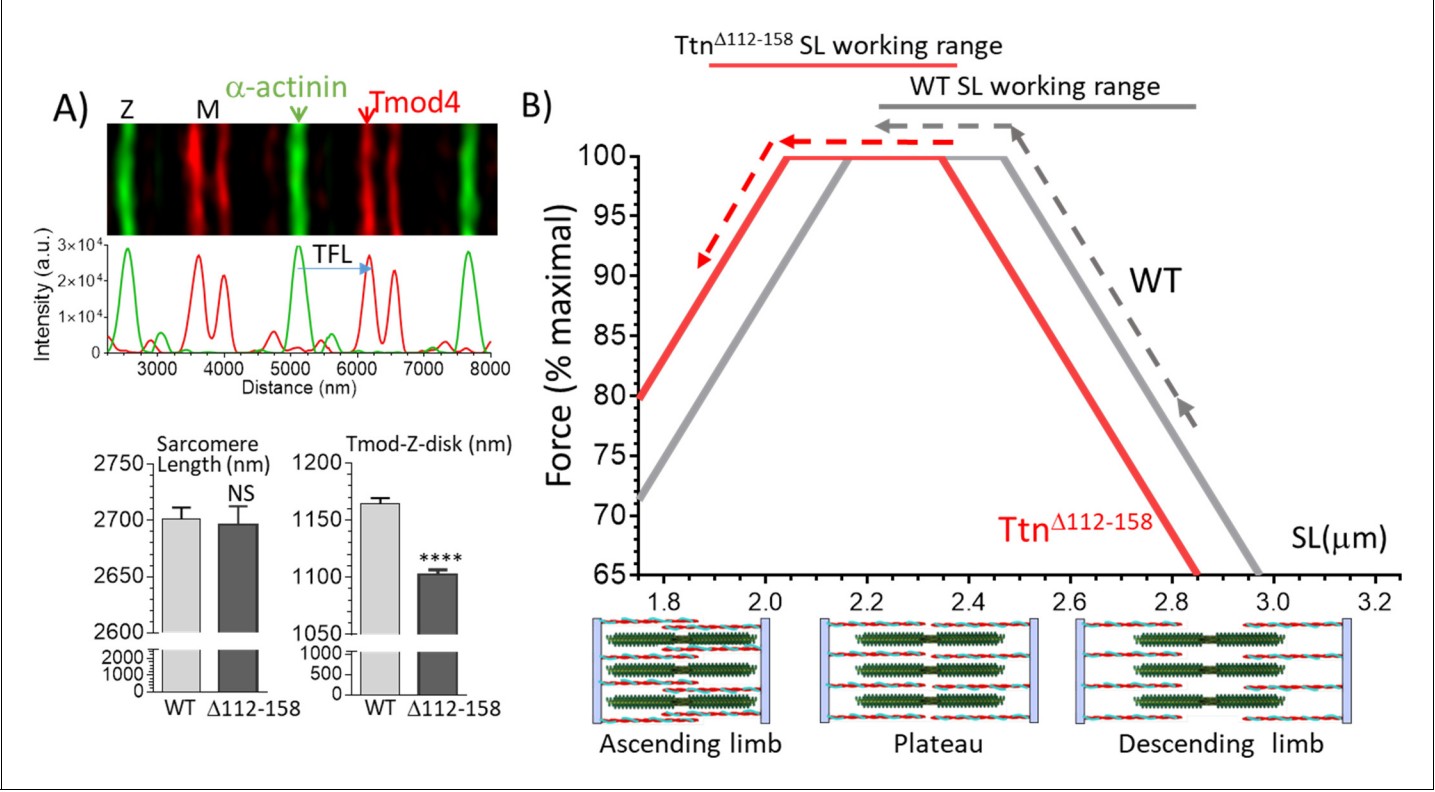

**Figure 11.** Thin filament length (TFL) and predicted force-sarcomere length (SL) relation. (**A**) TFL was measured in diaphragm muscle by super-resolution optical microscopy using the TFL-pointed end capping protein Tmod4. Top shows a WT example. Bottom: analyzed results. Sarcomere length (left) is not different. TFL (Tmod-Z-disk distance) is significantly reduced from 1165 nm in WT to 1102 nm in $Ttn^{\Delta112-158}$ diaphragm. (**B**) Top, Predicted force-sarcomere length relation using the measured TFLs: WT grey; $Ttn^{\Delta112-158}$ red. Broken lines indicate the sarcomere length working ranges of the WT and $Ttn^{\Delta112-158}$ diaphragm (from **Figure 8D**). (**B**) Bottom, schematic of sarcomeres on ascending limb (left), plateau (middle) and descending limb (right) of the force-SL relation. Green: thick filaments, red: thin filaments. (Note the central region of the thick filament that is devoid of myosin heads is known as the bare zone.).

DOI: https://doi.org/10.7554/eLife.40532.022

The following figure supplement is available for figure 11:

**Figure supplement 1.** Nebulin exon expression analysis in diaphragm muscle.

DOI: https://doi.org/10.7554/eLife.40532.023

## Passive muscle stiffness

Although it is well accepted that titin underlies the passive tension of the skeletal muscle single fiber (**Granzier and Wang, 1993**; **Granzier and Labeit, 2006**; **Wang et al., 1991**), titin's contribution to whole muscle stiffness, relative to that of the ECM, is not well established. Our mechanical studies on intact diaphragm, EDL and soleus muscles show that intact muscle passive tensions, of all muscle types studied, are much higher in $Ttn^{\Delta112-158}$ mice (**Figure 4**). These findings support that titin is a dominant passive tension source, unlike the conclusion drawn from earlier work that the ECM is dominant (**Gillies et al., 2011**; **Tirrell et al., 2012**). We also dissected the contribution of titin and the ECM to whole muscle passive stiffness using the EDL 5[th] toe muscle, a muscle chosen because it has as a major advantage that it is thin enough for sarcomere length measurement with laser diffraction. The muscles were studied first intact, then following skinning and a final time after KCl/KI extraction. The skinning vs. intact comparison shows that skinning has no effect on passive tension (**Figure 5A and B**) indicating that the sarcolemma and its direct connections to the ECM do not contribute to passive tension. This result is different from a study on fiber bundles isolated from rabbit muscle where skinning resulted in a 20–40% reduction in tension (**Prado et al., 2005**). It is not immediately obvious why the earlier result on rabbit fiber bundles differs from ours. We speculate that the dissection required to free the fiber bundles from their surrounding (including cutting the ends)

made the fiber bundles more vulnerable to skinning. The present work reveals that when using whole intact muscle, skinning does not affect passive tension.

The KCl/KI extraction that was used in the present work has previously been shown in cardiac muscle to remove titin's anchors in the sarcomere, thereby rendering titin mechanically inactive but leaving the ECM mechanically unaltered (*Granzier and Irving, 1995*; *Wu et al., 2000*). For this extraction method to be useful in skeletal muscle, the extraction needs to fully abolish all of titin's passive tension, while not affecting that of the ECM. To test whether this was the case, the KCl/KI sensitive tensions of skinned muscles were compared to the measured passive tensions of single fibers that had been mechanically skinned and that were, therefore, devoid of ECM. Identical results were obtained, in both WT and $Ttn^{\Delta112-158}$ mice (*Figure 5C*). This supports that the KCl/KI extraction specifically and fully abolishes titin's passive tension from whole muscle and that this extraction method is suitable to mechanically differentiate the titin-based passive tension (KCl/KI sensitive) from the ECM-based tension (KCl/KI insensitive). The obtained titin-based passive tension of muscle far exceeds that of the ECM and in the SL range used in this study (~2.2-3.2 μm) is 73 ± 4% of total tension in WT and 71 ± 6% in $Ttn^{\Delta112-158}$ muscle. Thus, titin is a dominant source of passive tension in both WT and $Ttn^{\Delta112-158}$ mice. The tension-sarcomere length data were also used to determine the average titin and ECM stiffness in the sarcomere length range of 2.45-2.75 μm. This range was selected because it captures most of the physiological sarcomere length range of the muscle types that were studied (*Figure 8D*), as well as that of many other muscle types in a wide range of species (*Burkholder et al., 1994*; *Burkholder and Lieber, 2001*). The stiffness of titin and ECM were both significantly increased in $Ttn^{\Delta112-158}$ EDL muscle but the titin stiffness far exceeded the ECM stiffness, ~5-fold in WT and ~10-fold in $Ttn^{\Delta112-158}$ muscle (*Figure 5D*). This does not mean that the ECM is unimportant for longitudinal stiffness. At sarcomere lengths at or beyond the physiological range limit, the ECM stiffness greatly increases and functions as a brake on the movement range of muscle. Within the physiological length range of skeletal muscle, however, titin's contribution to whole muscle stiffness is dominant. This is analogous to cardiac muscle where titin also dominates within the physiological sarcomere length (*Granzier and Irving, 1995*; *Wu et al., 2000*). An advantage of titin dominating passive stiffness is that this allows for rapid regulation through intracellular signaling mechanisms that post-translationally modify titin's molecular spring region (*Hidalgo and Granzier, 2013*).

## Active tension

In recent work, several novel mechanisms have been proposed in which titin affects the myosin-based active tension. Based on single-molecule studies on titin, it has been proposed that prior to activation, titin's immunoglobulin (Ig) domains unfold and that upon muscle activation these domains refold, adding to the developed force [26, 42]. Considering the higher passive forces in the $Ttn^{\Delta112-158}$ muscles, this mechanism predicts that unfolding of Ig domains will be more pronounced in $Ttn^{\Delta112-158}$ mice and that, consequently, the refolding-induced force of titin generated during activation will be higher, adding to the measured tetanic force level. However, the results of the present study (Fig. 6A and B) do not show higher active tensions in $Ttn^{\Delta112-158}$ muscles. The tetanic tension was measured at $L_0$, the experimentally determined length at which active tension is maximal (overlap between thin and thick filament during the tetanic tension plateau is optimal). At $L_0$, passive tension is significantly higher in $Ttn^{\Delta112-158}$ muscles than in WT muscles (for example passive tension in the EDL 5th toe muscle is 11.8±2 1.1 compared to 2.5 ±20.4 mN/mm² in WT muscle). Yet active tensions are not different (Fig. 6B). It is possible that the refolding mechanism is not active because passive forces at $L_0$ are too low, in both genotypes. Perhaps, at longer lengths sufficient levels of unfolding might take place, due to the higher passive tensions, and the mechanism is operational. Presently we conclude that at the macroscopic scale of a whole muscle contracting at $L_0$, the titin unfolding/refolding mechanism is unlikely to significantly add to the level of tetanic tension.

A second recent mechanism by which titin might affect active tension is based on the active state of the thick filament. It is well known that regulation of contraction in striated muscle is thin filament based, but new evidence suggests that the thick filament also has OFF and ON states (*Piazzesi et al., 2018*; *Fusi et al., 2016*). In the OFF state, many of the myosin heads in the thick filament are bent backward towards the center of the sarcomere, by an asymmetric arrangement of the two heads of the myosin molecule that folds the heads against the myosin tail (*Woodhead et al., 2005*). As a result, these heads are unable to interact with actin, that is they are OFF. In the ON

state, the myosin heads have moved away from the thick filament surface and have an orientation more perpendicular to the thick filament long axis, permitting interaction with the thin filament and generating force. It has been proposed that the ON state can be promoted by mechanically stressing the thick filament, either through the generation of active tension (by constitutively ON myosin molecules) or by titin-based passive tension (*Piazzesi et al., 2018*; *Fusi et al., 2016*). The molecular spring region of titin spans from near the Z-disk to the tip of the thick filament with the remaining segment of titin spanning along and interacting with the thick filament backbone (*Tonino et al., 2017*). Thus, when passive sarcomeres are stretched, titin stresses the thick filament and this promotes the ON state; upon calcium activation, this increased ON state then enhances force development (*Fusi et al., 2016*). The $Ttn^{\Delta112-158}$ model allows this activation model to be evaluated. Most relevant for the thick filament activation model results at submaximal activation because if titin-based tension sensitizes the system by enhancing the thick filament ON state, a measurable effect on force is most likely to occur under those conditions. However, no genotype effect was detected when examining the frequency required for half-maximal activation (*Figure 6C*). An increase in tension was seen in the EDL 5$^{th}$ toe muscle at frequencies < 60 Hz (average increase 15.7%) with a two-way ANOVA revealing a significant genotype effect ($p<0.001$), but this effect was not consistent in other muscle types. Interestingly, the time required to reach half-maximal tension was significantly reduced in all $Ttn^{\Delta112-158}$ muscle types, on average by 17.9% (*Figure 6D*). Because no fiber type switch was found in $Ttn^{\Delta112-158}$ muscles (*Figure 6—figure supplement 1*) it is unlikely that this increased speed of force development is due to the expression of faster fiber types. Instead, we speculate that this more rapid force development is due to the higher passive tension in $Ttn^{\Delta112-158}$ muscle that increases the thick filament ON state and that this accelerates force generation when a muscle is stimulated. Clearly, more work is required to examine the effect of titin-based tension on force generation of skeletal muscle and the $Ttn^{\Delta112-158}$ model will be useful for this purpose.

Finally, it has also been reported that the passive stiffness of titin increases in the presence of calcium. The effect is likely due to glutamate rich (E-rich) domains within the PEVK region of titin that have a reduced persistence length in the presence of calcium (*Labeit et al., 2003*). This gives rise to a 10–20% increase in passive stiffness in the presence of calcium, the level of which varies with the number of E-rich domains that are expressed in the titin isoforms found in different muscle types (*Labeit et al., 2003*; *Fujita et al., 2004*). The calcium sensitivity of E-rich PEVK domains might also contribute to the force enhancement that occurs after an active muscle has been stretched (*Herzog, 2005*). It is noteworthy that most E-rich PEVK domains that are expressed in skeletal muscle are located in the N-terminal end of the PEVK element that has been deleted in the $Ttn^{\Delta112-158}$ mouse (*Granzier et al., 2007*). Thus, the $Ttn^{\Delta112-158}$ mouse model will be very useful for future studies of the role of E-rich domains in muscle physiology.

## Sarcomere length working range

In all examined muscle types of $Ttn^{\Delta112-158}$ mice, a large shift was found in the physiological working range, towards shorter sarcomere lengths (*Figure 8*). That this shift is functionally important can be evaluated by plotting the average passive stiffness of titin in the physiological sarcomere length range. This reveals an identical passive stiffness in the two genotypes (see *Figure 12*). Thus, the adaptation in the physiological sarcomere length range of the $Ttn^{\Delta112-158}$ mice is remarkably effective at normalizing passive stiffness. This adaptation might explain why exercise performance of $Ttn^{\Delta112-158}$ mice is normal (*Figure 7*). The shifted sarcomere length working range is expected to have a minimal impact on active tension development because of the reduced thin filament length that was found in $Ttn^{\Delta112-158}$ mice that left shifted the force-sarcomere length relation (*Figure 11*). Considering that nebulin regulates thin filament length (*Kruger et al., 1991*), we examined whether nebulin in $Ttn^{\Delta112-158}$ mice has a reduced size but no differences were detected in nebulin expression (*Figure 11—figure supplement 1*). Hence it is unlikely that nebulin is involved in the thin filament length reduction of $Ttn^{\Delta112-158}$ mice. Considering that the thin filament contains a distal nebulin-free segment (*Littlefield and Fowler, 2008*), we speculate that in $Ttn^{\Delta112-158}$ mice this nebulin-free distal segment is reduced in length, in response to the shift in the sarcomere length working range.

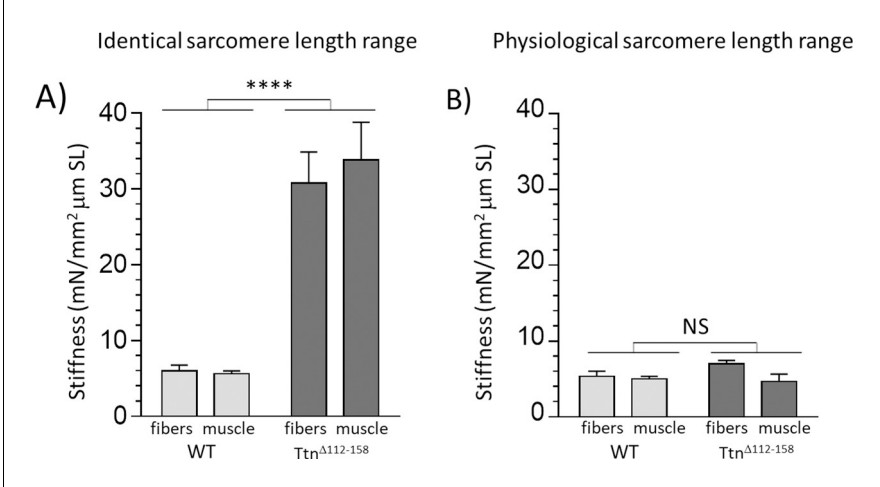

**Figure 12.** Passive muscle stiffness at the physiological sarcomere length range. Passive stiffness in $Ttn^{\Delta112-158}$ and WT EDL 5[th] toe muscle at an identical sarcomere length range (**A**) and at the physiological sarcomere length of each genotype (**B**). Data in A represent the average stiffness in the sarcomere length range 2.45–2.75 μm (data from **Figure 5C**, inset) and data in B represent the average stiffness in the sarcomere length range 2.35–2.85 μm for WT and 2.12–2.55 μm for $Ttn^{\Delta112-158}$, representing the physiological sarcomere length range determined in **Figure 8D**. It is notable that within the physiological sarcomere length range (**B**) passive stiffness of the two genotypes is the same. (No differences between results of single fibers and whole muscle.). Legends for the supplementary figures.

DOI: https://doi.org/10.7554/eLife.40532.024

## Sarcomeres in series and longitudinal hypertrophy

The shift in the working sarcomere length range in $Ttn^{\Delta112-158}$ mice can be explained by the increase in the number of sarcomeres in series that was found (**Figure 9**). Adaptations in the number of sarcomeres in series are well known to occur in response to maintaining a muscle in either a lengthened state (the number of series sarcomeres increases) or in a shortened state (the number of series sarcomeres decreases) (**Williams and Goldspink, 1978**; **Tabary et al., 1972**). To our knowledge, our study is the first in which the number of sarcomeres is increased primarily in response to increasing the stiffness of titin without changing muscle length. It is possible that the adaptation in sarcomere number based on lengthening or shortening muscle and the adaptation due to an increased passive stiffness of titin share a common mechanism. The mechano-signaling mechanism identified recently that involves an increased strain on titin's molecular spring as well as the upregulation of titin-binding proteins (**van der Pijl et al., 2018**) is a good candidate. However, an expression analysis of mechanosensitive proteins including titin-binding proteins did not reveal consistent differences between WT and $Ttn^{\Delta112-158}$ muscles (results not shown). A subsequent RNAseq-based expression analysis using WT and $Ttn^{\Delta112-158}$ Diaphragm, EDL, and soleus muscles did reveal multiple differentially expressed genes, but follow-up protein expression studies did not support their relevance (results not shown). We speculate that the underlying signaling mechanisms that cause longitudinal hypertrophy are only transiently active and that these mechanisms are 'dialed down' when the increase in the number of sarcomeres in series has been accomplished (and titin's strain has been normalized). A transient activation is also suggested by the longitudinal muscle hypertrophy in $Ttn^{\Delta112-158}$ mice that is independent of age (most muscle types in **Figure 10** and **Figure 10—figure supplement 1** show no significant interaction between genotype and age). Clearly, future work is needed to elucidate the mechanisms that underlie the serial addition of sarcomeres. This work should focus on changes that occur at an early age when the longitudinal growth that increases the number of series sarcomeres is not yet complete.

## Summary

Within the physiological sarcomere length range of skeletal muscle, titin is the main determinant of skeletal muscle passive stiffness. Shortening titin's molecular spring region in $Ttn^{\Delta112-158}$ mice results

in longitudinal hypertrophy that increases the number of sarcomeres in series and, importantly, that shifts the in vivo sarcomere length working range to shorter lengths where operating passive stiffness is low. The expected negative effect of this shift on active force generation is minimized through shortening of thin filaments. The lowering of passive stiffness through the serial addition of sarcomeres requires multiple major adaptations, indicating that increased passive stiffness is functionally detrimental and has to be avoided. Normalizing function through the addition of ~30% sarcomeres is energetically costly, as suggested by the reduced weight of the $Ttn^{\Delta112-158}$ mice (*Figure 1D*), and ultimately might be detrimental.

## Materials and methods

### Key resources table

| Reagent type (species) or resource | Designation | Source or reference | Identifiers | Additional information |
|---|---|---|---|---|
| gene (Mus musculus) | Titin (Ttn) | NA | Ensembl: ENSMUSG00000051747 | |
| genetic reagent (Mus musculus) | $Ttn^{\Delta112-158}$ | this paper | | Deletion of (GRCm38/mm10) chr2:76,839,202–76,867,333 |
| strain, strain background (Mus musculus) | 'mixed C57BL/6J and 129P2/OlaHsd " | other | | Homologous recombination in E14TG2a cells (129/Ola strain, PMID: 3683574). Mice backcrossed seven generations to C57BL/6J (JAX stock#664) |
| antibody | anti-titin N2A (chicken polyclonal) | Biogenes | Biogenes: anti-X105-X106 | (5.75 µg/mL) |
| antibody | anti-titin I84-86 (rabbit polyclonal) | Biogenes | Biogenes: anti-BK283-BK284 | (1 µg/mL) |
| antibody | anti-α-actinin (mouse monoclonal) | Sigma-Aldrich | Sigma-Aldrich:A7811; RRID:AB_476766 | (1:1000) |
| antibody | anti-Tmod4 (rabbit polyclonal) | Proteintech Group Inc | Proteintech Group:11753–1-AP; RRID:AB_2205433 | (1:750) |
| antibody | AlexaFluor-488 conjugated goat anti-mouse IgG | Invitrogen | | (1:500) |
| antibody | AlexaFluor-568 conjugated goat anti-rabbit IgG | Invitrogen | | (1:250) |
| antibody | AlexaFluor-647 conjugated goat anti-mouse IgG | Invitrogen | | (1:250) |
| antibody | CF-633 conjugated donkey anti-chicken IgY | Biotium | | (1:200) |
| other | AlexaFluor-488 conjugated phalloidin | Invitrogen | | (1:1000) |

### Mice

All experiments in this study complied with the NIH Guide for the Care and Use of Laboratory Animals and were approved by the University of Arizona's IACUC (Institutional Animal Care and Use Committee). The Genetically Engineered Mouse Model Core (University of Arizona, Tucson, AZ, USA) produced the $Ttn^{\Delta112-158}$ mice by homologous recombination in $Hprt^{-/-}$ E14TG2a#25 embryonic stem cells derived from strain 129/Ola (*Doetschman et al., 1987*). The targeting vector was assembled into a vector including a pMC1 neocassette bordered by loxP sites. The $Ttn^{\Delta112-158}$ mice were backcrossed with C57BL/6J mice for seven generations. The $Ttn^{\Delta112-158}$ deletion corresponds to a deletion of 47 PEVK exons, encoding 1586 amino acids or 181.7 kDa.

## Body weight analysis and muscle dissection

Mice were placed under isoflurane anesthesia before performing a cervical dislocation. Body weights were measured and the heart, the diaphragm, and the skeletal muscles of the lower limbs were dissected and weighed; tibia length (TL) were also measured. The muscles were frozen in liquid $N_2$. Muscle weights were collected at 60 days, 80 days, 6 months, and 9 months of age.

## Intact muscle mechanics on diaphragm, soleus and whole EDL muscle

After isoflurane anesthesia and cervical dislocation, either a strip of the left costal of the diaphragm, the EDL, or the soleus was quickly dissected from the mouse and placed into oxygenated Ringer's solution. Sutures were then tied around both tendons of the muscle and the muscle was mounted in a heated bath (30°C) filled with Ringer's solution with 95%/5% $O_2$/$CO_2$ bubbled through, with one suture attached to a fixed end and the other to a force transducer/length controller (Aurora Scientific Model 300C-LR-FP). After letting the muscle equilibrate, the muscle was stretched to its optimal length ($L_0$) of maximal force output using 20 Hz stimulations. Then a 150 Hz tetanus was performed to tighten the sutures before starting the protocols. First, a passive stretch protocol was performed. The muscle was stretched in steps 5% for the diaphragm, 3.5% for the soleus and 2.5% for the EDL of the muscle's slack length at 1% per second followed by a 20 s hold to allow for force relaxation (*Ottenheijm and Granzier, 2010*). For the diaphragm, the muscle was stretched by a total of 50% of muscle length while the EDL and soleus muscles were stretched by 25% and 35%, respectively (The EDL and Soleus have fibers that only run part of the way of the whole muscle, so the fiber length to muscle length ratio has to be considered when estimating sarcomere length; *Burkholder et al., 1994*). The fiber length to muscle length ratios were attained from the whole-body formaldehyde perfusion study (see below). The diaphragm has parallel fibers going from end to end of the muscle and therefore did not need to be adjusted. After allowing the muscle to reach a steady state at the end of the 20 s hold, force was determined.

Following the passive stretch protocol there is a 7 min waiting period, and then $L_0$ was reestablished and a force-frequency protocol was performed. The force-frequency protocol involves stimulating the muscle at varying frequencies using electrodes submerged in the bath. From this data, the maximal force produced, the minimal force produced, the time it takes to reach maximal force, the time the muscle takes to relax, and the frequency required to reach ½ of the maximal force can be extrapolated by fitting the force-frequency curve. The force-frequency curve was fit using the sigmoidal equation: $P_0(F) = P_{0min} + \left( P_{0max} - P_{0min} / \left\{ 1 + exp\left[\frac{F_{half}-F}{k}\right] \right\} \right)$ (*Prosser et al., 2011*) where $P_{0min}$ gives the minimum specific force, $P_{0max}$ gives the maximum specific force, $F_{half}$ defines the frequency where $P_0 = 0.5$ of $P_{0max}$, and $1/k$ is a measure of the steepness of the $P_0$ vs. F relationship. The curves for the different genotypes were also tested for significance using an extra sum of squares F-test. Passives stiffness curves were also compared by fitting a logarithmic curve to the data and comparing the curves using an extra sum of squares F-test. The force-frequency and passive stretch forces produced by the EDL and Soleus were normalized by the physiological cross-sectional area (PCSA) of the muscle. The PCSA of the EDL and Soleus muscles were determined by using the measured muscle mass, muscle length, and taking the measured pennation angle of the fibers and the fiber length to muscle length ratio into account (*Lieber and Ward, 2011*). The PCSA was calculated as (*Lieber and Ward, 2011*): $PCSA\ (cm^2) = \frac{muscle\ mass\ (g) * cos(\theta)}{\rho (g\ cm^{-3}) * fiber\ length\ (cm)}$ ($\theta$ is the pennation angle and $\rho$ is the physiological density of muscle). The PCSA used for normalizing forces produced by the diaphragm was determined from the width and thickness (determined optically using a calibrated eyepiece) and assuming an elliptical shape of the cross-section of the muscle strip. Following the force-frequency protocol, the muscle was removed from the bath and pinned at its slack length and fixed for at least an hour using 3% paraformaldehyde and 2% glutaraldehyde in PBS (phosphate-buffered saline). Fiber bundles were then dissected and the sarcomere lengths were measured using high-magnification optical microscopy (Aurora Scientific Model 901A HVSL, Model 600A v1.81 Digital Controller). From this sarcomere length, the sarcomere lengths during the passive stretch protocol were determined using the equation: SL + ((%Stretch/100)/fiber length:muscle length ratio) x SL); SL = Sarcomere Length obtained at the slack muscle length. The fiber length-muscle length ratio in both WT and *Ttn*$^{\Delta 112-158}$ EDL and soleus muscle was found to be identical and consistent with the

literature (for WT mouse muscle), that is 0.71 for soleus and 0.51 for EDL muscle (*Burkholder et al., 1994*).

## EDL 5th Toe muscle mechanics-intact, skinned and extracted muscle

The EDL 5th toe muscle was dissected for intact mechanics and mounted on the Aurora Scientific 1500A small isolated muscle system with the proximal suture attached to a force transducer (Aurora Scientific Models 402A) and the distal end to a high-speed length controller (Aurora Scientific Models 322C-I). Prior to starting the protocols, a150Hz tetanus was performed under modest passive tension to settle the preparation. The optimal length for active force generation ($L_0$) was determined using 1 Hz twitch stimulations. The muscle length was measured using a calibrated eyepiece and the SL was measured in real time using laser diffraction from a class 3B laser (Milles Griot 25-LHR-991–249). A force frequency protocol was performed as described above. The muscle was then set to slack length and a 30% passive stretch protocol was implemented. The muscle was removed from the setup and skinned overnight in 1% Triton X-100 in relaxing solution(in mM: 40 BES, 10 EGTA, 6.56 MgCl2, 5.88Na-ATP, 46.35 K-propionate, 15 Creatine Phosphate, 1 DTT, 1 E64, 1 Leupeptint, 1.25 PMSF). The following day, the muscle was washed with relaxing solution then re-mounted on the setup in relaxing solution maintained at 20°C. SL length was confirmed at $L_0$, then the muscle was returned to slack length and the 30% passive stretch protocol repeated. Titin anchor extraction was then performed using a high KCl and then a high KI solution in-situ for two 30 min incubations each (*Littlefield and Fowler, 2008*). The solutions contained 0.6 mM of KCl and 1 mM of KI respectively (*Wu et al., 2000*). To ensure the complete detachment of titin from the Z-disk, a passive stretch protocol was performed immediately following the 2nd replacement with high KI solution. After the completion of the final 30 min KI incubation, the passive stretch protocol was repeated. SLs measured after skinning were used for analysis. The forces produced by the 5th toe EDL were normalized by the PCSA based on the weight of the muscle measured immediately after dissection, an 11.3% pennation angle, and a fiber length ratio of 0.69 (*Chleboun et al., 1997*). Force-frequency and passive stretch forces were normalized with PCSA calculated using $L_0$ and slack length respectively.

## Single fiber mechanics

EDL 5th toe muscles were dissected and skinned overnight in 1% Triton X-100 in relaxing solution (in mM: 40 BES, 10 EGTA, 6.56 MgCl2, 5.88Na-ATP, 46.35 K-propionate, 15 Creatine Phosphate, 1 DTT, 1 E64, 1 Leupeptint, 1.25 PMSF). The following day, the muscles were washed thoroughly with relaxing solution and single fibers were isolated from the 5th toe EDL muscle. In addition to the chemical skinning, mechanical skinning was performed using fine-tipped forceps to remove the endomysium around every single fiber. Small aluminum clips were attached to the mechanically skinned fiber at each end. The fiber was hooked up between a force transducer (ASI 403A, Aurora Scientific Inc.) and a length motor (ASI 322C, Aurora Scientific Inc., Ontario, Canada) in a skinned fiber apparatus (ASI 802D, Aurora Scientific Inc.) that was mounted on an inverted microscope (IX73, Olympus Inc, PA, USA). Sarcomere length was measured using a high-speed VSL camera and ASI 900B software (Aurora Scientific Inc.). The fiber was set to slack length (the shortest length at which passive force first develops) and the diameter and depth of the fiber were measured at four points along the fiber. The cross-sectional area of the fiber was calculated assuming an elliptical shape. The tension was calculated by dividing force with fiber cross-sectional area. The fiber was stretched a total of 49% of slack length, in 7% steps, at a rate of 1% per second, followed by a 20 s hold at the end of each step.

## RNA sequencing (RNAseq)

EDL, Soleus, and diaphragm samples were collected from 60-dayold male mice and placed into RNA later in order to preserve RNA integrity. We studied per muscle type 8 WT and 8 $Ttn^{\Delta112-158}$ muscles (from eight animals) and pooled two muscles providing four samples per muscle type and genotype for RNAseq. For RNA extraction, 600 µl pre-chilled buffer RLT (RNeasy Fibrous Tissue Mini Kit, Qiagen) with 1% β-Mercaptoethanol was added to muscle tissue stored in RNAlater in a 4 ml cryovial. Tissue was disrupted using a rotor-stator-homogenizer for 30 s. A protein digest was performed by adding 600 µl RNase-free water containing 6 mAU Proteinase K and incubating at 55°C

for 10 min. Samples were transferred to a 1.5 ml microfuge tube and centrifuged for 3 min at 14000 g. The supernatant was transferred to a 2 ml tube with 600 µl Ethanol and transferred to an RNeasy mini spin column. Thereafter, RNA extraction was performed following the manufacturer's instructions and quantified using a Nanodrop ND-1000 spectrophotometer (Thermo Scientific).

For the library preparation, rRNA was depleted from RNA preparations with a NEBnext rRNA depletion kit using 1 µg total RNA as starting material. Libraries were prepared using the NEBNext Ultra II Directional RNA Library Prep Kit for Illumina following the manufacturer's instructions. RNA was fragmented for 10 min at 94°C. For first strand cDNA synthesis, incubations were performed for 10 min at 25°C followed by 50 min at 42°C and 15 min at 70°C. For size selection, conditions for an approximate insert size of 300 bp were used. Size-selected libraries were enriched by PCR for 10 cycles and purified using NEBnext sample purification beads. Library quality was checked using a fragment analyzer and sequencing performed on an Illumina Hiseq2500 sequencer using 100 bp paired-end sequencing.

Adapters and low-quality reads were removed with Trim Galore (http://www.bioinformatics.babraham.ac.uk/projects/trim_galore/) and reads were mapped to the mouse genome (Release M17 GRCm38.p6) using STAR (*Dobin et al., 2013*) with default settings. Differentially expressed genes were determined with DESeq2 (*Love et al., 2014*). Genes with adjusted p-values lower than 0.05 were considered to be differentially expressed. Selected differentially expressed genes were analyzed in $Ttn^{\Delta112-158}$ samples using western blots to check if changes in gene expression correlate with altered protein levels.

For calculating the inclusion percentage of all exons from titin and nebulin transcripts, inclusion reads and exclusion reads were counted for each exon based on annotation from Bang *et al.* (*Bang et al., 2001*) for titin and Kazmierski *et al.*(*Kazmierski et al., 2003*) for nebulin. Inclusion reads (IR) are reads overlapping the exon being investigated, normalized by exon length. Exclusion reads (ER) are reads either upstream or downstream that support exclusions of the read. From these factors the following equations were used to calculate the percent spliced in index (PSI) as a measure of much an exon is spliced in:

$$IR_{i,n} = \frac{IR_i}{length\ exon_i + read\ length - 1}$$

$$ER_{i,n} = \frac{ER_i}{read\ length - 1}$$

$$PSI_i = \frac{IR_{i,n}}{IR_{i,n} + ER_{i,n}}\%$$

Where *i* is the exon number and *n* is the normalized read counts. Determination of differential exon usage was performed after adjusting exon counts to gene counts by:

$$E_{ijk}^A = \frac{E_{ijk} \times \bar{G_j}}{G_{jk}}$$

Where $E_{ijk}^A$ is the adjusted exon count for exon *i* of gene *j* in sample *k*, $E_{ijk}$ is the raw exon count for exon *i* of gene *j* in sample *k*. $\bar{G_j}$ is the mean raw count for gene *j* for all samples and $G_{jk}$ is the raw gene count for gene *j* in sample *k*. Adjusted exon counts were then used as input for a statistical analysis identical to the method provided for analysis of differential gene expression provided by the edgeR package[59].

## MHC isoform analysis

MHC isoform analysis was performed as previously described (*Li et al., 2015*). Briefly, Myosin heavy chain isoform composition was visualized using 8% acrylamide gels stained with Coomassie blue. MHC type I and IIB are well separated on gels but the IIA overlaps with a small amount of IIX that exists in mouse skeletal muscle. Therefore, we refer to this band as IIA/X.

## Formaldehyde perfusion study and ultrasound

We followed the method explained in Van der Pijl (*van der Pijl et al., 2018*). 60 day old mice were sedated using a ketamine/xylazine cocktail. Once sedated, the mice were cannulated in the left jugular vein that was securely tied off. Using an echo machine, the left costal of the diaphragm was visualized at the height of the vena cava. A B-mode image was recorded to establish strain on the diaphragm. One of the lower limbs of the mouse was fixed in a fully plantar flexed position while the other leg was fixed in a fully dorsiflexed position, taking care not to force the joints into a non-physiological position. Before injection of formaldehyde, hair on the abdomen was removed using NAIR cream (Church and Dwight Co.) and ultrasound data were collected. The probe was put on the abdomen of the mouse at a 45° angle aiming to the thoracic cavity. Standard B-mode images of the diaphragm were obtained and were analyzed using Vevo Strain software (Fujifilm VisualSonics; Toronto, Canada). A single respiration cycle was selected using a reiterated M-mode image. Using a free curve, a 4–5-point region of interest was set in the arching region of the costal diaphragm dome to measure strain. The inferior vena cava was used as a reference point to ensure data was taken from the same area in each mouse.

After cutting the femoral vein, 3 mL 4% formaldehyde in phosphate-buffered saline (PBS) was injected through the cannula at a constant rate of 0.1 mL/s. An M-mode image was collected during diaphragm arrest. After waiting for five additional minutes for the fixation to take place, the whole diaphragm was dissected out while still attached to the ribcage, and the diaphragm with ribcage was placed into a 4% formaldehyde in PBS solution and fixed overnight. The two limbs of the mouse were removed by cutting the limbs at the femur and the entire leg was placed into a 4% formaldehyde in PBS overnight. The diaphragm, both EDLs, and both Soleus muscles were carefully dissected in PBS. The length of the diaphragm fibers of the left costal were measured using a calibrated eyepiece. The left costal was then dissected into small bundles and sarcomere length was measured using laser-diffraction. The EDL and Soleus muscles from both limbs were digested in 15% sulfuric acid for 40 min before fibers were carefully dissected, ensuring full lengths of the fibers were kept intact. The lengths of these fibers were then measured using a calibrated eyepiece. Following this measurement, the sarcomere length of the EDL and Soleus was measured using laser-diffraction. Using the fiber lengths and the total muscle length, a fiber-length to muscle-length ratio was calculated. From the sarcomere length and the fiber lengths, the number of sarcomeres in series was calculated.

## Free wheel running

Mice were placed individually in a cage containing a running wheel with no resistance when the mice were 40 days of age. These mice could run voluntarily on the running wheel. Daily exercise values for running distance, time, and speed were collected (Lafayette Instruments Model 80820FS). The mice continued to exercise at will for 40 days. Data measured in 5 day-running intervals starting at day one when the running wheel was presented. The values in *Figure 7A* are the average of days 31–35. After 40 days the mice were sacrificed and muscles were harvested and muscle weights collected.

## Metabolic treadmill

Mice ran on an indirect calorimetry treadmill (TSE Systems). Mice underwent a graded exercise tolerance test until exhaustion using the following parameters (speed, duration, incline grade): (0 m/min, 3 min, 0°), (6 m/min, 2 min, 0°), (9 m/min, 2 min, 5°), (12 m/min, 2 min, 10°), (15 m/min, 2 min, 15°), (18 m/min, 1 min, 15°), (21 m/min, 1 min, 15°), (23 m/min, 1 min, 15°), and (+1 m/min, each 1 min thereafter, 15°). Exhaustion was defined as the point at which mice maintained continuous contact with the shock grid for 5 s or 20 visits to the shock grid in a 1 min period. During testing, gas was collected continuously and analyzed every 5 s. PhenoMaster software (TSE Systems) recorded and calculated oxygen consumption (VO2), carbon dioxide consumption (VCO2), the respiratory exchange ratio (RER), distance ran, speed ran, duration, and visits to the shock grid. VO2max was determined by the peak oxygen consumption reached during this test when RER was >1.0. Maximum running speed was defined as the treadmill speed at which VO2max was achieved. Mice failing to reach an RER >1.0 were excluded.

## Titin protein expression analysis

Titin isoforms expression was determined as described previously (*Hidalgo et al., 2014*). Briefly, solubilized muscle samples were electrophoresed on 1% agarose gels. Each gel was stained with Coomassie blue, scanned, and analyzed densitometrically. The slope of the linear relationship of titin isoforms and MHC was determined to quantify expression ratios. We analyzed separately T1 titin (the full-length molecule), T2 titin (a major titin degradation product) and total titin (T1 and T2 combined). The same protocol was used in order to investigate possible nebulin size changes. For these experiments WT, $Ttn^{\Delta112-158}$, and two samples which contain heterozygous expression of both WT nebulin and a smaller or larger nebulin. The mobility of nebulin in the $Ttn^{\Delta112-158}$ samples was compared for the heterozygous nebulin samples to get insight into possible nebulin size changes in the $Ttn^{\Delta112-158}$ samples.

## Super-resolution structured illumination microscopy (SR-SIM)

The immunofluorescence experiments were performed as previously described (*Tonino et al., 2017*). Dissection of muscles was performed mice anesthetized with isoflurane (USP, Phoenix Pharmaceuticals, Inc.) and sacrificed by cervical dislocation. A diaphragm strip from the left costal and the EDL muscle were quickly dissected and skinned in relaxing solution ([in mM]: 40 BES, 10 EGTA, 6.56 MgCl2, 5.88 Na-ATP, 1 DTT, 46.35 K-propionate, 15 creatine phosphate, pH 7.0) containing 1% (w/v) Triton X-100 and protease inhibitors ([in mM]: 0.01 E64, 0.047 leupeptin and 0.25 PMSF). The diaphragm strip and EDL muscle were skinned overnight at 4°C, washed thoroughly with relaxing solution and stored in 50% glycerol/relaxing solution at −20°C and used within two weeks for experiments. Female mice (n = 4 in each group) aged around 60 days were used in the study. Skinned left costal diaphragm and EDL fiber bundles were embedded in O.C.T compound and immediately frozen in 2-methylbutane precooled in liquid nitrogen. 4 μm thick cryosections were then cut and mounted onto microscope slides. Tissue sections were permeabilized in 0.2% Triton X-100/PBS for 20 min at room temperature, blocked with 2% BSA and 1% normal donkey serum in PBS for 1 hr at 4°C, and incubated overnight at 4°C with primary antibodies diluted in blocking solution. The primary antibodies included: a chicken polyclonal anti-X105-X106 (N2A) (5.75 μg/mL) (Myomedix), a rabbit polyclonal anti-BK283-BK284 (I84-86) (1 μg/mL) (Myomedix), a mouse monoclonal anti-α-actinin (1:1000) (EA-53, Sigma-Aldrich) antibody and a rabbit polyclonal anti-Tmod4 (1:750) (Proteintech Group). Sections were then washed with PBS for 2 × 30 min and incubated with secondary antibodies diluted in PBS for 3 hr at room temperature. The secondary antibodies included: AlexaFluor-488 conjugated goat anti-mouse IgG (1:500, Invitrogen), AlexaFluor-568 conjugated goat anti-rabbit IgG (1:250, Invitrogen), AlexaFluor-647 conjugated goat anti-mouse IgG (1:250, Invitrogen), CF-633 conjugated donkey anti-chicken IgY (1:200, Biotium).

The sections were then washed with PBS for 2 × 15 min and covered with number 1.5H coverslips using ProLong Diamond mounting medium (Thermo Scientific, Inc.). A Zeiss ELYRA S1 SR-SIM microscope was used with UV and solid-state laser (488/561/642 nm) illumination sources, a 100 × oil immersion objective (NA = 1.46), and a sCMOS camera. Typical imaging was performed on a 49.34 × 24.67 μm² area with 1280 × 640 pixel dimensions. Image stacks comprising of 40 slices were acquired with 0.101 μm Z-steps, five angles and five phases/angles for each slice. Image reconstruction and fluorescence intensity plot profile generation were performed with ZEN two software (Zeiss). Plot profiles were fitted with Gaussian curves to determine the epitope peak position using Fityk 1.3.0 software. Note that the N2A and I84-86 antibodies do not bind exactly adjacent to the PEVK and this was accounted for by subtracting 15 nm (~4 Ig domains) from the measured values, to determine the extension of the PEVK, z. Thin filament length was determined by measuring the distance between Tmod epitopes across a Z-disk and dividing by two (i.e., the length contains the half Z-disk width.)

The measured thin filament length and the 1.6 μm thick filament length and 0.15 μm bare-zone width from the literature (*Sosa et al., 1994*) allows the construction of the force-SL relation in WT and $Ttn^{\Delta112-158}$ diaphragm as follows. Force is maximal along the plateau of the force-sarcomere length relation where the thin filament tip is within the bare-zone of the A-band (see *Figure 11B*, middle bottom panel). At longer sarcomere lengths where the thin filaments do not reach the bare-zone, the sarcomeres are on the descending limb (see also *Figure 11B*, right bottom) and force decreases linearly with sarcomere length (−69% of maximal force per μm sarcomere length) to reach

zero force at the sarcomere length where overlap is zero (two times thin filament length plus thick filament length). Finally, at short sarcomere lengths where the thin filament tip is beyond the edge of the bare zone in the adjacent half sarcomere, sarcomeres are on the ascending limb (see also *Figure 11B*, left bottom) and force increases linearly with sarcomere length. The submaximal force on the ascending limb is due to the force inhibition in the thin filament overlap zone; the magnitude of the slope of the ascending limb is +69% of maximal force per µm sarcomere length. For details see (*Granzier and Pollack, 1990*; *Granzier et al., 1991*).

### WLC modeling

To evaluate the effects of the shortened PEVK segment on titin-based stiffness we used the experimentally obtained extension (z) of the PEVK segment (above). The contour length (CL) of the PEVK segment was estimated from the number of amino acids contained in the segment (see above) and assuming a random coil structure with a maximal residue spacing of 0.38 nm (*Watanabe et al., 2002b*), the relative extension of the PEVK was calculated as z/CL. The obtained values were used in the wormlike chain (WLC) force equation and the force per titin molecule was determined. The WLC equation: $F \times PL/k_B \times T = z/CL + 1/(4(1 - z/CL)^2) - 1/4$ (*Kellermayer et al., 1997*; *Bouchiat et al., 1999*). The WLC model describes the molecule as a deformable continuum of persistence length PL (a measure of bending rigidity); $k_B$ is Boltzmann's constant and T is absolute temperature. The PL was taken as 1.4 nm (*Labeit et al., 2003*). For fitting, we used Levenberg-Marquardt nonlinear fits of the WLC model (written in IgorPro (Wavemetrics) and KaleidaGraph(Synergy software)).

### Statistical analysis

Data are reported as means ± standard error. Findings were labeled as significant if the p-value was found to be under 0.05 and then labeled as '*'. A $p <= 0.01$ is shown as '**' and $p <= 0.005$ is shown as '***'. One or two-way ANOVA analyses were used to establish differences between the two genotypes when taken as a group. Whenever practical both male and female mice were used (no sex differences were found in any of our studies).

## Acknowledgments

Research reported in this publication was supported by the National Institute of Arthritis and Musculoskeletal and Skin Diseases of the National Institutes of Health under Award Number R01AR053897 and R01AR073179. We are grateful to Ms. Luan Wyly, Mr. Chandra Saripalli, Ms. Xiaoqun Zhou, Mr. Xiangdang Liu, Ms. Zaynab Hourani and Mr. Odhin Brynnel for important support of this work.

## Additional information

### Funding

| Funder | Grant reference number | Author |
| --- | --- | --- |
| National Institute of Arthritis and Musculoskeletal and Skin Diseases | R01AR053897 | Henk L Granzier |
| National Institute of Arthritis and Musculoskeletal and Skin Diseases | R01AR073179 | Henk L Granzier |

The funders had no role in study design, data collection and interpretation, or the decision to submit the work for publication.

### Author contributions

Ambjorn Brynnel, Conceptualization, Resources, Data curation, Formal analysis, Supervision, Funding acquisition, Validation, Investigation, Methodology, Writing—original draft, Project administration, Writing—review and editing; Yaeren Hernandez, Data curation, Investigation, Methodology, Writing—original draft; Balazs Kiss, Maya Adler, Data curation, Formal analysis; Johan Lindqvist, Data curation, Formal analysis, Methodology; Justin Kolb, Robbert van der Pijl, Joshua Strom, John Smith,

Coen Ottenheijm, Data curation; Jochen Gohlke, Conceptualization, Data curation; Henk L Granzier, Conceptualization

### Author ORCIDs
Henk L Granzier http://orcid.org/0000-0002-9516-407X

### Ethics
Animal experimentation: This study was performed in strict accordance with the recommendations in the Guide for the Care and Use of Laboratory Animals of the National Institutes of Health. All of the animals were handled according to approved institutional animal care and use committee (IACUC) protocols (#09-095) of the University of Arizona.

### Decision letter and Author response
Decision letter https://doi.org/10.7554/eLife.40532.027
Author response https://doi.org/10.7554/eLife.40532.028

## Additional files

### Supplementary files
• Transparent reporting form
DOI: https://doi.org/10.7554/eLife.40532.025

### Data availability
All data generated or analysed during this study are included in the manuscript and supporting files. Source data files have been provided for Figure 2 and corresponding supplements (please see Figure 2-source data 1).

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
