## [Decision Letter]

[**Editorial note:** This article has been through an editorial process in which the authors decide how to respond to the issues raised during peer review. The Reviewing Editor's assessment is that all the issues have been addressed.]

Thank you for submitting your article "Downsizing the giant titin reveals its dominant roles in skeletal muscle passive stiffness and longitudinal hypertrophy" for consideration by *eLife*. Your article has been reviewed by three peer reviewers, and the evaluation has been overseen by a Reviewing Editor and Arup Chakraborty as the Senior Editor. The following individuals involved in review of your submission have agreed to reveal their identity: Jorge Alegre-Cebollada (Reviewer #1); Walter Herzog (Reviewer #2); Julio Fernandez (Reviewer #3).

The Reviewing Editor has highlighted the concerns that require revision and/or responses, and we have included the separate reviews below for your consideration. If you have any questions, please do not hesitate to contact us.

The authors deleted an extremely large stretch of titin that is believed to be responsible for passive elasticity, and mice with that deletion indeed turn out to have a reduction in passive stiffness. The mice themselves are surprisingly normal, likely due to compensating changes in the muscle. The study is very relevant and experiments are expertly performed.

The manuscript can benefit from a revised discussion ion how the data helps us understand the role of titin in active contraction. For example, Reviewer 3 suggests, based on prior literature, the compensatory changes occur to main the relevant mechanical tension across titin.

Separate reviews (please respond to each point):

*Reviewer #1:*

In the study by Brynnel and collaborators, the authors generate and characterize very thoroughly a mouse model (Delta^112-158^) in which the PEVK region of titin is genetically shortened to assess the role of the protein in skeletal muscle passive stiffness. The topic is highly relevant since titin is a main player in the mechanical performance of striated muscle, and because different diseases originate from mutations in the titin gene. Hence, there is a pressing need in the field to understand how the mechanical properties of titin molecules influence the performance of striated muscle. The group has produced before several mouse models in which different segments of titin have been altered. The novelty of the Delta^112-158^ model comes from the fact that it involves a massive deletion (46 exons) in the PEVK region of the protein, which is believed to be one of the major contributors to the passive mechanical properties of titin. It is very interesting to examine how mice adapt to such a massive deletion in titin, which the authors do in their manuscript combining several top-notch physiology and molecular biology techniques.

The authors provide data supporting the expression of shortened titin in the homozygous Delta^112-158^ mice, both at the RNA and protein levels. Using super-resolution microscopy on muscle preparations and modelling, the authors confirm that Delta^112-158^ titin is stiffer than wild-type, as expected. Interestingly, even though higher muscle stiffness is found in Delta^112-158^ mice, the authors find that mice do well in general terms (including exercise tests). This increased stiffness is found at the level of intact, skinned and KCl/KI-extracted muscle. Indeed, results are in agreement with a prominent role of titin in setting the passive stiffness of skeletal muscle, since the stiffness of Delta^112-158^ muscles correlates very well with the increase in titin-based stiffness. Titin-based stiffness is deduced from data obtained after KCl/KI extraction, a method that can potentially affect other muscle components. However, they provide nice supporting data obtained with single fibers, which are naturally devoid of extracellular matrix (Figure 5C). The authors also measured active contraction, and found that Delta^112-158^ mice develop force faster than wild-type mice, with no difference in the maximal force developed.

In our opinion the most striking observation is that there are compensatory changes that enable myocytes to adjust to the shortened titin, e.g. reduced sarcomere length working range and shortened thin filaments, which are accompanied by longitudinal hypertrophy at the muscle level. These adaptations probably lead to normal viability and development of Delta^112-158^ mice.

Although the characterization of the new model is very thorough already, we believe that there are highly interesting questions to address in the future, such as how is the new sarcomere length working rate set?, how early in development these adjustments occur?, are these adjustments already completed at the embryo stage? It may be interesting also to analyze heterozygous mice to check how myocytes adapt to a titin stiffness imbalance.

We have some concerns regarding the current version of the manuscript:

1) It seems that adaptation in sarcomere length working range can lead to equal working stiffness of titin (and muscle) in wild-type and Delta^112-158^ mice, which would be a very interesting observation. Can the authors compare stiffness at the sarcomere length working range for both wild-type and Delta^112-158^ mice? The insets in Figures 3C, 4, and 5 are calculated at the same sarcomere lengths, which can be misleading since sarcomere length working ranges are different for both mice.

2) It is argued in subsection “Active tension” that results with Delta^112-158^ mice provide evidence against the recently proposed mechanisms of titin folding contributing to active contraction. In our opinion, the arguments raised by the authors do not support this view because they don't take into account the different sarcomere length working ranges (see above). Results in Figure 3C suggest that at the different sarcomere length working ranges (as measured in Figure 7D), both WT and Delta^112-158^ titin experience very similar forces and hence the expected rate of folding/unfolding in both cases should be very similar.

Minor Comments:

3) The facts that Delta^112-158^ mice weigh less than wild-type, but have heavier muscles are somehow contradictory. Can the authors provide an explanation? Do Delta^112-158^ mice have less adipose tissue, less water content, etc? This should be at least discussed and more data provided if already available.

4) It would be nice to describe in the Materials and methods section how the force-sarcomere length relations were calculated (Figure 10B).

5) Some techniques employed by the authors are very specialized. Although the Materials and method section is in general very detailed, it may help to show some illustrative raw data in some cases, for instance the kind of images used to calculate CSA.

6) Subsection “Force-sarcomere length relation”: A-band length instead of width

7) Subsection “Sarcomeres in series and longitudinal hypertrophy”: what does "interaction between genotype and age" refer to? Consider rephrasing.

*Reviewer #2:*

This is an excellent and comprehensive evaluation of the changes in muscle structure and function in a genetic model, Ttn^112-158^, of 46 exons deletions in titin's spring region. By shortening titin's spring region, titin became significantly stiffer, and affected passive force (interestingly not active force) in fibers and muscles, and was associated with structural changes (other than just the changes to titin) most remarkably, a shortening of the actin filaments by about 63nm, an increase in fiber lengths, and a vast increase (30%) in serial sarcomeres.

Some small limitations of the study include that:

i) titin stiffness was never measured directly, but was obtained indirectly. Titin stiffness or passive force contribution as a function of sarcomere length can be directly obtained in single myofibrils, thus eliminating any concerns of indirect measurements where titin is deleted from muscles/fibers, and skinning may affect (although apparently not in this preparation), the passive forces.

ii) it has been argued by that titin's stiffness changes/increases upon activation and actin-myosin-based force production. The idea that titin might play a different role in activated muscle compared to passive muscle was not discussed, and with some of the authors having published evidence of a change in titin stiffness with activation/force production/muscle stretch, this might have added to a more complete discussion.

iii) ranges of sarcomere excursions in muscles were derived from fixed (passive) specimens. It is accepted that depending on the muscle and the length of the muscle, fibers and average sarcomere lengths decrease between 5-30% in active compared to passive muscles. If we assume that the Ttn^112-158^ muscles have greater passive stiffness, then shortening of sarcomeres in these muscles upon activation might be less than for the wild-type muscles. Therefore, the sarcomere excursions between the Ttn^112-158^ and WT muscles might be closer during active force production than the passive results suggest.

Minor Comments:

The only minor comment I have is regarding a statement in the Introduction section. Here it is said that: "…. and in cerebral palsy patients, titin-based passive stiffness is increased several fold [22, 23]". Reference 22 is not dealing with cerebral palsy, and reference 23 has just one reference to cerebral palsy (Friden and Lieber, 2003). Measurements in that study were performed in single fibers of patients with cerebral palsy, and they found that the elastic modulus of the passive fibers was doubled compared to control. However, fibers do not only give titin stiffness and passive force contributions but an unknown amount of passive force can also come from other sources, which is hard to control and estimate in single fibers.

We just performed a study on single myofibrils from spastic muscles of patients with cerebral palsy and found that single myofibril passive forces (at given sarcomere lengths) and stiffness (at given sarcomere lengths) were about 50% smaller than those of corresponding myofibrils from control subjects. These passive forces are exclusively due to titin in myofibrils. Measuring titin molecular mass, we did not find a difference between the control and cerebral palsy samples. However, when quantifying the amount of titin, we found that the amount of titin was reduced by about 50% in the samples from the cerebral palsy patients compared to control. Unfortunately, our study is still in the review process, and thus does not need to be considered here. But likely, titin-based stiffness, as directly assessed by us, rather than the indirect methods used by Friden and Lieber, 2003, seems much less in patients with cerebral palsy than controls because of a reduction in the amount of titin.

Additional data files and statistical comments:

There is no need (from my point of view) to add additional files and/or to change any of the statistical approaches. They seem perfectly appropriate.

*Reviewer #3:*

Granzier and colleagues have a long history of designing and characterizing mouse models in which the titin gene has been altered. The phenomenology that emerges from their work is useful in our budding attempts to understand how titin is integrated into the structure of muscle. Their previous models have included deletion of the elastic N2B segment from myofibrils, M-line domains, splicing factors responsible for regulating titin length, and a piece of the PEVK segment. This last paper ("Truncation of titin's elastic PEVK region leads to cardiomyopathy with diastolic dysfunction", Circulation Research, 2009) identified a marked cardiac phenotype and also established that titin was the principal source of muscle elasticity and not its collagen sheath. In the current work, by removing more exons to truncate the PEVK segment by 1180 amino acids, they get new phenotypes, this time in skeletal muscles. While deletion of the C-terminal exons of the PEVK segment (exons 219-225) resulted in a diastolic dysfunction phenotype with increased titin based passive tension (Circ Res, 2009), deletion of the N-terminal PEVK segment here (exons 112-158) caused a very different phenotype where they now observe a shift in several different muscle types, to shorter sarcomere length, shorter thin filaments, and longitudinal hypertrophy where an increased number of sarcomeres spans the same muscle length. It is likely that these adaptations result from a homeostatic change aimed at keeping the mean force per titin at a constant working value. This can be easily pictured by drawing a horizontal line, say at 5 pN, in both panels of Figure 3C and finding the intercepts with the WT and TtnD122-158 mice. It is clear then that to keep the titin working force at the same set-point, the mutant mouse has to work at sarcomere lengths that are shorter by about 0.5 micrometers which is exactly what the authors observe in vivo.

Why is it that the organism places such a premium on keeping the force per titin at a constant value? A likely explanation can be found in Rivas-Pardo et al., 2016 and Eckels et al., 2018 who proposed that titin folding plays a role both in muscle elasticity and contraction. This theory demands that titin operates precisely over a range of forces that straddles the folding probability of the titin Ig domains (~4-8 pN): If the force is too high (>12 pN), folding never occurs, and if it is too low titin is always folded (< 3 pN). In either case, titin folding would not play a role. By contrast, if titin operates in vivo at forces that are on average in the middle of the Ig domain folding probability curve, say ~5-6 pN, then muscle tissue can take advantage of the storage and release of elastic energy that takes place with Ig domain folding. If this were true, muscle physiology would place a high premium on getting titin to operate over a tight range of forces. Indeed, Rivas-Pardo et al., predicted that the optimal sarcomere length in vivo would closely track this fundamental requirement adjusting widely amongst different muscle types, tracking their alternatively spliced differences in stiffness (Figures 5C and 5D in Rivas Pardo et al., 2016).

Unwittingly, the current work by Granzier and colleagues lends strong support to this hypothesis. How the titin operating force is transduced into such large scale adaptations is a work in progress. However, we do know that titin has a force sensor at its C-terminus end (Mayans Nature 1998, Lange Science 2005) and that undergoes force-induced structural changes to activate downstream pathways of gene expression, presumably as a feedback mechanism to control the operating force range. Thus, a coherent view emerges, where a high premium is placed by the organism on exactly controlling the operating force of titin molecules; this has the signature of protein folding under force.

More excitingly, the current work offers an approach for a definitive test of this theory. If all the Ig domains of the proximal I band region were replaced by PEVK segments -while maintaining the same molecular weight-, then titin would be converted into a mostly passive entropic spring. The kinase control would adjust the sarcomere length to restore the correct operating force in all muscles, and the sarcomere would become longer to achieve the correct resting tension per titin, however, there would be no Ig domain folding/unfolding reactions. If under these conditions the peak contractile force remains unchanged, then the mechanical power developed by Ig domain folding plays no role in muscle contraction. However, If Ig domain folding is crucial for muscle contraction, then obviously the peak force would be greatly diminished by the removal of the Ig domains and the experiments would validate the hypothesis. As they stand, the current experiments do not answer this question. Given that the authors have removed only the elastomeric PEVK segment but left the Ig domains intact, the Ig folding mechanisms remains a viable theory.

---

## [Author Response]

The authors deleted an extremely large stretch of titin that is believed to be responsible for passive elasticity, and mice with that deletion indeed turn out to have a reduction in passive stiffness. The mice themselves are surprisingly normal, likely due to compensating changes in the muscle. The study is very relevant and experiments are expertly performed.The manuscript can benefit from a revised discussion ion how the data helps us understand the role of titin in active contraction. For example, Reviewer 3 suggests, based on prior literature, the compensatory changes occur to main the relevant mechanical tension across titin.

We have expanded the discussion of the role of titin in active contraction. For details, please see below.

Separate reviews (please respond to each point):

Reviewer #1:

[…] In our opinion the most striking observation is that there are compensatory changes that enable myocytes to adjust to the shortened titin, e.g. reduced sarcomere length working range and shortened thin filaments, which are accompanied by longitudinal hypertrophy at the muscle level. These adaptations probably lead to normal viability and development of Delta^112-158^ mice.

We agree that the compensatory changes are striking and that they highlight that titin-based stiffness is functionally important and that its level has to be controlled.

Although the characterization of the new model is very thorough already, we believe that there are highly interesting questions to address in the future, such as how is the new sarcomere length working rate set?, how early in development these adjustments occur?, are these adjustments already completed at the embryo stage? It may be interesting also to analyze heterozygous mice to check how myocytes adapt to a titin stiffness imbalance.

These interesting research questions are currently being addressed in our ongoing research. Developmental studies are challenging and time-consuming and we hope to report their results during the next year.

We have some concerns regarding the current version of the manuscript:1) It seems that adaptation in sarcomere length working range can lead to equal working stiffness of titin (and muscle) in wild-type and Delta^112-158^ mice, which would be a very interesting observation. Can the authors compare stiffness at the sarcomere length working range for both wild-type and Delta^112-158^ mice? The insets in Figures 3C, 4, and 5 are calculated at the same sarcomere lengths, which can be misleading since sarcomere length working ranges are different for both mice.

To address this great suggestion, existing data from the EDL 5^th^ toe muscle were analyzed. We selected this muscle type because its sarcomere length can be directly measured with laser diffraction, making the requested comparison relatively straightforward. We evaluated the titin-based stiffness in whole muscle (KCl/KI sensitive tension of skinned whole muscle) and in single fibers. The two genotypes were compared at the same sarcomere length range and, importantly, at their experimentally determined physiological sarcomere length working range. At the same sarcomere length range, average passive stiffness is much higher in the Ttn^Δ112-158^EDL muscles (Figure 7A) but at their physiological sarcomere length ranges, stiffness in the Ttn^Δ112-158^EDL muscle is identical to that of WT muscle (newly added Figure 7B). Thus, the shift in the sarcomere length working range of the Ttn^Δ112-158^ mice (towards shorter lengths) is remarkably effective in normalizing passive stiffness. We highlighted this new result in the revised manuscript, subsection “Sarcomere length working range”.

2) It is argued in subsection “Active tension” that results with Delta^112-158^ mice provide evidence against the recently proposed mechanisms of titin folding contributing to active contraction. In our opinion, the arguments raised by the authors do not support this view because they don't take into account the different sarcomere length working ranges (see above). Results in Figure 3C suggest that at the different sarcomere length working ranges (as measured in Figure 7D), both WT and Delta^112-158^ titin experience very similar forces and hence the expected rate of folding/unfolding in both cases should be very similar.

Active tensions were not measured at their physiological sarcomere lengths as this would likely result in the two genotypes demonstrating a tetanic force difference due to unequal overlap between thin and thick filaments. Instead, tetanic force was measured at L_0_, i.e., the optimal muscle length for tetanic force generation (L_0_ is commonly used as a reference length in intact muscle experiments as it results in tetanic force generation at optimal overlap between thin and thick filaments). The passive tension level at L_0_ was higher in Ttn^Δ112-158^ muscle, for example in the EDL 5th toe muscle passive tension was 11.8 ± 1.1 (n=9) compared to 2.5 ± 0.4 mN/mm2 (n=9) in WT muscle. The higher passive tension in Ttn^Δ112-158^ muscle relative to WT muscle is expected to cause a higher degree of unfolding prior to activation. As a result, internal sarcomere shortening occurring during the tetanus (see also below) is expected to cause a higher degree of refolding in Ttn^Δ112-158^ muscle and, according to the recently proposed titin refolding mechanism, tetanic tension is expected to be higher in Ttn^Δ112-158^ muscle. Yet measured active tensions are not different (Figure 6A and B). In the revised manuscript we expanded our discussion of this issue that you raised, including highlighting that our conclusions only apply to force measurements at L_0_. Please see subsection “Active tension”.

Minor Comments:3) The facts that Delta^112-158^ mice weigh less than wild-type, but have heavier muscles are somehow contradictory. Can the authors provide an explanation? Do Delta^112-158^ mice have less adipose tissue, less water content, etc? This should be at least discussed and more data provided if already available.

The implication of the body weight (lower), muscle weight (higher) and body size (the same, based on tibia length) data is that the Ttn^Δ112-158^mice are leaner due to less body fat tissue. As suggested we now discuss this topic in paragraph two of subsection “The number of sarcomeres in series and longitudinal muscle growth”.

4) It would be nice to describe in the Materials and methods section how the force-sarcomere length relations were calculated (Figure 10B).

Apologies for not having described this well enough. We have expanded out explanation in the Materials and methods section. See the final paragraph of subsection “Super-resolution Structured Illumination Microscopy (SR-SIM)”.

5) Some techniques employed by the authors are very specialized. Although the Materials and method section is in general very detailed, it may help to show some illustrative raw data in some cases, for instance the kind of images used to calculate CSA.

For the diaphragm muscles we measured with an optical microscope the width and depth of the muscle strips and the CSA was calculated by assuming that the cross-sectional area is elliptical. For the EDL and soleus muscles, we calculated the physiological cross-sectional area (PCSA) of the muscle using the measured muscle mass, muscle length, the pennation angle of the fibers, and the fiber length to muscle length ratio (this is not needed for the diaphragm because fibers run parallel and from end-to-end of the muscle strip). The PCSA of the EDL and soleus were calculated using the following equation: PCSA(cm2)=musclemassg*cos⁡θ⁡ρgcm-3*fiberlengthcm (Lieber and Ward, 2011; θ is the pennation angle and ρ is the physiological density of muscle). Thus there are no CSA images to show but instead we have expanded our explanation of the CSA measurement in the Materials and methods section: second paragraph of subsection “Intact Muscle Mechanics on Diaphragm, soleus and whole EDL muscle”.

6) Subsection “Force-sarcomere length relation”: A-band length instead of width

A-band width is what is usually used in the muscle literature but you are right that this is confusing and we changed to ‘length’.

7) Subsection “Sarcomeres in series and longitudinal hypertrophy”: what does "interaction between genotype and age" refer to? Consider rephrasing.

This refers to the statistical analysis that was used.In our study, muscle weights were measured at four different ages (60, 80, 183, and 274 days of life) and in two genotypes (WT and Ttn^Δ112-158^). The muscle weight data were analyzed with a two-way ANOVA with age and genotype as factors. This revealed that in most muscle types both genotype and age had a significant effect on muscle weight (i.e., muscles were hypertrophied in Ttn^Δ112-158^mice and hypertrophy increased with age). Two-way ANOVA analysis also reveals the interaction between the factors that are evaluated, which is of interest as it will reveal whether hypertrophy in Ttn^Δ112-158^mice is progressive (which is what we initially expected), or is independent of age. For most muscle types we found no significant interaction, indicating that age similarly affected WT and Ttn^Δ112-158^muscle weights. In the revised manuscript we better explain what is meant by interaction. Please see subsection “The number of sarcomeres in series and longitudinal muscle growth”.

Reviewer #2:

This is an excellent and comprehensive evaluation of the changes in muscle structure and function in a genetic model, Ttn^112-158^, of 46 exons deletions in titin's spring region. By shortening titin's spring region, titin became significantly stiffer, and affected passive force (interestingly not active force) in fibers and muscles, and was associated with structural changes (other than just the changes to titin) most remarkably, a shortening of the actin filaments by about 63nm, an increase in fiber lengths, and a vast increase (30%) in serial sarcomeres.Some small limitations of the study include that:i) titin stiffness was never measured directly, but was obtained indirectly. Titin stiffness or passive force contribution as a function of sarcomere length can be directly obtained in single myofibrils, thus eliminating any concerns of indirect measurements where titin is deleted from muscles/fibers, and skinning may affect (although apparently not in this preparation), the passive forces.

A major goal of the present study was to better understand the importance of titin for whole muscle function and, therefore, many experiments were focused at the whole muscle level. However, we did perform comparative studies on mechanically skinned muscle fibers, a preparation with passive stiffness that is solely due to titin (as long as extreme sarcomere lengths are avoided (>~4.0 μm) where intermediate filaments significantly contribute, see Granzier and Wang, *Biophys J*. 1993;65(5):2141-59). Although myofibrils could have been used for this work, we believe that the single fiber preparation works equally well. An advantage is that compared to myofibrils, the muscle fiber has a structural organization that is closer to that of intact muscle.

ii) it has been argued by that titin's stiffness changes/increases upon activation and actin-myosin-based force production. The idea that titin might play a different role in activated muscle compared to passive muscle was not discussed, and with some of the authors having published evidence of a change in titin stiffness with activation/force production/muscle stretch, this might have added to a more complete discussion.

We wholeheartedlyagree and have added this as a discussion topic. See revised manuscript, final paragraph of subsection “Active tension”.

iii) ranges of sarcomere excursions in muscles were derived from fixed (passive) specimens. It is accepted that depending on the muscle and the length of the muscle, fibers and average sarcomere lengths decrease between 5-30% in active compared to passive muscles. If we assume that the Ttn^112-158^ muscles have greater passive stiffness, then shortening of sarcomeres in these muscles upon activation might be less than for the wild-type muscles. Therefore, the sarcomere excursions between the Ttn^112-158^ and WT muscles might be closer during active force production than the passive results suggest.

In previous work on single muscle fibers we controlled sarcomere length using a feedback system and prevented sarcomere shortening during contraction. Unfortunately sarcomere length control has not been achieved at the whole muscle level, due to, for example, the sarcomere length signal that is too noisy. Instead we limited series compliance bytying muscles to the force transducer very close to their tendon insertions. In the EDL 5^th^ toe muscle we were able to follow sarcomere length during activation in some muscles. Minimal internal shortening was observed (typically ~5-10% ) without an indication that internal shortening was less in the Ttn^Δ112-158^muscles. It is also relevant to note that the sarcomere length range of the diaphragm was based on a strain analysis on contracting muscle, using ultrasound imaging (Figure 7A). This revealed that the diaphragm strain amplitude (fractional shortening of the muscle fibers during contraction) did not vary between Ttn^Δ112-158^and WT mice.

Minor Comments:The only minor comment I have is regarding a statement in the Introduction section. Here it is said that: "…. and in cerebral palsy patients, titin-based passive stiffness is increased several fold [22, 23]". Reference 22 is not dealing with cerebral palsy, and reference 23 has just one reference to cerebral palsy (Friden and Lieber, 2003). Measurements in that study were performed in single fibers of patients with cerebral palsy, and they found that the elastic modulus of the passive fibers was doubled compared to control. However, fibers do not only give titin stiffness and passive force contributions but an unknown amount of passive force can also come from other sources, which is hard to control and estimate in single fibers.We just performed a study on single myofibrils from spastic muscles of patients with cerebral palsy and found that single myofibril passive forces (at given sarcomere lengths) and stiffness (at given sarcomere lengths) were about 50% smaller than those of corresponding myofibrils from control subjects. These passive forces are exclusively due to titin in myofibrils. Measuring titin molecular mass, we did not find a difference between the control and cerebral palsy samples. However, when quantifying the amount of titin, we found that the amount of titin was reduced by about 50% in the samples from the cerebral palsy patients compared to control. Unfortunately, our study is still in the review process, and thus does not need to be considered here. But likely, titin-based stiffness, as directly assessed by us, rather than the indirect methods used by Friden and Lieber, 2003, seems much less in patients with cerebral palsy than controls because of a reduction in the amount of titin.

Thank you for pointing this out.Citations 22 and 23 referred back to both FSHD and cerebral palsy and to avoid confusion we have now moved citation 22 immediately after the FSHD discussion. We have also softened our statement with respect to titin in cerebral palsy. See revised Introduction section. The work that you discussed is interesting and we look forward to learning more about it and including it in our discussions of future papers.

Additional data files and statistical comments:There is no need (from my point of view) to add additional files and/or to change any of the statistical approaches. They seem perfectly appropriate.

Reviewer #3:

*[…] While deletion of the C-terminal exons of the PEVK segment (exons 219-225) resulted in a diastolic dysfunction phenotype with increased titin based passive tension (Circ Res, 2009), deletion of the N-terminal PEVK segment here (exons 112-158) caused a very different phenotype where they now observe a shift in several different muscle types, to shorter sarcomere length, shorter thin filaments, and longitudinal hypertrophy where an increased number of sarcomeres spans the same muscle length. It is likely that these adaptations result from a homeostatic change aimed at keeping the mean force per titin at a constant working value. This can be easily pictured by drawing a horizontal line, say at 5 pN, in both panels of Figure 3C and finding the intercepts with the WT and TtnD122-158 mice. It is clear then that to keep the titin working force at the same set-point, the mutant mouse has to work at sarcomere lengths that are shorter by about 0.5 micrometers which is exactly what the authors observe* in vivo.

Thank you for reviewing our manuscript and your thoughtful and constructive comments. We would like to briefly highlight that most of the previous models were created in order to study the role of titin in the heart, for example the model in which exon 49 was deleted is a cardiac-specific model that does not affect skeletal muscle titins (exon 49 is not included in skeletal muscle titin). In contrast, the Ttn^Δ112-158^mouse model is the first model with an expected skeletal muscle specific phenotype as the targeted exons are not expressed in the main cardiac isoform (the N2B titin isoform).

*Why is it that the organism places such a premium on keeping the force per titin at a constant value? A likely explanation can be found in Rivas-Pardo et al., 2016 and Eckels et al., 2018 who proposed that titin folding plays a role both in muscle elasticity and contraction. This theory demands that titin operates precisely over a range of forces that straddles the folding probability of the titin Ig domains (~4-8 pN): If the force is too high (>12 pN), folding never occurs, and if it is too low titin is always folded (< 3 pN). In either case, titin folding would not play a role. By contrast, if titin operates* in vivo *at forces that are on average in the middle of the Ig domain folding probability curve, say ~5-6 pN, then muscle tissue can take advantage of the storage and release of elastic energy that takes place with Ig domain folding. If this were true, muscle physiology would place a high premium on getting titin to operate over a tight range of forces. Indeed, Rivas-Pardo et al., predicted that the optimal sarcomere length* in vivo would closely track this fundamental requirement adjusting widely amongst different muscle types, tracking their alternatively spliced differences in stiffness (Figures 5C and 5D in Rivas Pardo et al., 2016).Unwittingly, the current work by Granzier and colleagues lends strong support to this hypothesis. How the titin operating force is transduced into such large scale adaptations is a work in progress. However, we do know that titin has a force sensor at its C-terminus end (Mayans Nature 1998, Lange Science 2005) and that undergoes force-induced structural changes to activate downstream pathways of gene expression, presumably as a feedback mechanism to control the operating force range. Thus, a coherent view emerges, where a high premium is placed by the organism on exactly controlling the operating force of titin molecules; this has the signature of protein folding under force.

We agree that the adaptation in the sarcomere length working range in the Ttn^Δ112-158^model is striking and that as a consequence the operating force levels per titin molecule are normalized. As per Figure 3C and 7D the operating force levels per titin molecule are in both WT and in Ttn^Δ112-158^mice below ~3 pN, possibly explaining why the folding mechanism does not augment tetanic tension (see above). It is worth noting that when tetanizing skeletal muscle on the descending limb (lengths in excess of L_0_ and with passive tension towards the limit of the physiological range), tetanic tension exceeds the level expected based on the degree of overlap between thin and thick filaments (this is best seen in the work of ter Keurs, Iwazumi and Pollack, *J Gen Physiol.* 1978;72(4):565-92). Perhaps the titin folding mechanism plays a role under those conditions, a notion consistent with the finding that when maintaining sarcomere length constant during tetanic contractions on the descending limb (using a sarcomere length feedback system), the tetanic tensions are reduced and are as expected from the degree of overlap between thin and thick filaments (e.g., Granzier and Pollack, *J Physiol*. 1990 Feb;421:595-615.) In our revised manuscript we discuss the topic that you raised, see subsection “Active tension”.

More excitingly, the current work offers an approach for a definitive test of this theory. If all the Ig domains of the proximal I band region were replaced by PEVK segments -while maintaining the same molecular weight-, then titin would be converted into a mostly passive entropic spring. The kinase control would adjust the sarcomere length to restore the correct operating force in all muscles, and the sarcomere would become longer to achieve the correct resting tension per titin, however, there would be no Ig domain folding/unfolding reactions. If under these conditions the peak contractile force remains unchanged, then the mechanical power developed by Ig domain folding plays no role in muscle contraction. However, If Ig domain folding is crucial for muscle contraction, then obviously the peak force would be greatly diminished by the removal of the Ig domains and the experiments would validate the hypothesis. As they stand, the current experiments do not answer this question. Given that the authors have removed only the elastomeric PEVK segment but left the Ig domains intact, the Ig folding mechanisms remains a viable theory.

Thank you for your thought-provoking comments.Unfortunately, the mouse model that you describe would be extremely difficult to make due to the extreme size of the targeted region (~80 kb), which far exceeds what is currently possible with either CRISPR-Cas9 or homologous recombination techniques. Additionally the Ig domains of the proximal tandem Ig are not continuous but have intermingled several novel exons, as well as the cardiac-specific N2B exon 49 (see Bang et al., 2001). Thus, multiple models would have to be made in which different regions of the proximal tandem Ig would have to be targeted sequentially. Presently we can conclude from the Ttn^Δ112-158^model that making titin stiffer does not alter the maximal tetanic force at the optimal muscle length (L_0_). However, it would be interesting to evaluate whether the Ig folding mechanism contributes to active tension at lengths longer than L_0_ where the forces on titin are higher and unfolding during stretch and refolding during contraction might be more prominent. In the revised manuscript we address this possibility. Please see subsection “Active tension”.